

**Algorithm Theoretical Baseline for formaldehyde retrievals**
**from S5P TROPOMI and from the QA4ECV project.**
Isabelle De Smedt[1], Nicolas Theys[1], Huan Yu[1], Thomas Danckaert[1], Christophe Lerot[1],
Steven Compernolle[1], Michel Van Roozendael[1], Andreas Richter[2], Andreas Hilboll[2], Enno
Peters[2], Mattia Pedergnana[3], Diego Loyola[3], Steffen Beirle[4], Thomas Wagner[4], Henk Eskes[5],
Jos van Geffen[5], Klaas Folkert Boersma[5,6], Peepijn Veefkind[5].
[1]{Royal Belgian Institute for Space Aeronomy (BIRA-IASB), Brussels, Belgium}
[2]{Institute of Environmental Physics, University of Bremen (IUP-B), Otto-Hahn-Allee 1, 28359 Bremen, Germany}
[3]{Institut für Methodik der Fernerkundung (IMF), Deutsches Zentrum für Luft und Raumfahrt (DLR), Oberpfaffenhofen, Germany}
[4]{ Max Planck Institute for Chemistry (MPIC), Hahn-Meitner-Weg 1, 55128 Mainz, Germany}
[5]{KNMI, De Bilt, The Netherlands}
[6]{Wageningen University, Meteorology and Air Quality group, Wageningen, The Netherlands}
*Correspondence to*: I. De Smedt (isabelle.desmedt@aeronomie.be)
Abstract: On board of the Copernicus Sentinel-5 Precursor (S5P) platform, the TROPOspheric Monitoring
Instrument (TROPOMI) is a double channel nadir-viewing grating spectrometer measuring solar back-
scattered earthshine radiances in the ultraviolet, visible, near-infrared and shortwave infrared with global daily
coverage. In the ultraviolet range, its spectral resolution and radiometric performance are equivalent to those
of its predecessor OMI, but its horizontal resolution at true nadir is improved by an order of magnitude. This
paper introduces the formaldehyde (HCHO) tropospheric vertical column retrieval algorithm implemented in
the S5P operational processor, and comprehensively describes its various retrieval steps. Furthermore,
algorithmic improvements developed in the framework of the EU FP7-project QA4ECV are described for
future updates of the processor. Detailed error estimates are discussed in the light of Copernicus user
requirements and needs for validation are highlighted. Finally, verification results based on the application of
the algorithm to OMI measurements are presented, demonstrating the performances expected for TROPOMI.
**1. Introduction**
Long term satellite observations of tropospheric formaldehyde (HCHO) are essential to support air quality and
chemistry-climate related studies from the regional to the global scale. Formaldehyde is an intermediate gas in
almost all oxidation chains of non-methane volatile organic compounds (NMVOC), leading eventually to $CO_2$.
NMVOCs are, together with $NO_x$, CO and $CH_4$, among the most important precursors of tropospheric ozone.
NMVOCs also produce secondary organic aerosols and influence the concentrations of OH, the main
tropospheric oxidant. The major HCHO source in the remote atmosphere is $CH_4$ oxidation. Over the continents,
the oxidation of higher NMVOCs emitted from vegetation, fires, traffic and industrial sources results in
important and localised enhancements of the HCHO levels (as illustrated in Figure 1). Its lifetime being of the
order of a few hours, HCHO in the boundary layer can be related to the release of short-lived hydrocarbons,
which mostly cannot be observed directly from space. Furthermore, HCHO observations provide information
on the chemical oxidation processes in the atmosphere, including CO chemical production from $CH_4$ and
NMVOCs. The seasonal and inter-annual variations of the formaldehyde distribution are principally related to



temperature changes (controlling vegetation emissions) and fire events, but also to changes in anthropogenic
activities. For all these reasons, HCHO satellite observations are used in combination with tropospheric
chemistry transport models to constrain NMVOC emission inventories in so-called top-down inversion
approaches (e.g. Abbot et al., 2003, Palmer et al., 2006; Fu et al., 2007; Millet et al., 2008; Stavrakou et al.,
2009a, 2009b, 2012, 2015; Curci et al., 2010; Barkley et al., 2011, 2013: Fortems-Cheiney et al., 2012; Marais
et al., 2012; Mahajan et al., 2015).
HCHO tropospheric columns have been successively retrieved from GOME on ERS-2 and from SCIAMACHY
on ENVISAT, resulting in a continuous data set covering a period of almost 16 years from 1996 until 2012
(Chance et al., 2000; Palmer et al., 2001; Wittrock et al., 2006; Marbach et al., 2009; De Smedt et al., 2008;
2010). Started in 2007, the measurements made by the three GOME-2 instruments (EUMETSAT METOP-A,
B and C) have the potential to extend by more than a decade the successful time-series of global formaldehyde
morning observations (Vrekoussis et al., 2010; De Smedt et al., 2012; Hewson et al., 2012; Hassinen et al.,
2016). Since its launch in 2004, OMI on the NASA AURA platform has been providing complementary HCHO
measurements in the early afternoon with daily global coverage and a better spatial resolution than current
morning sensors (Kurosu et al., 2008; Millet et al., 2008; González Abad et al., 2015; De Smedt et al., 2015).
TROPOMI aims to continue this time series of early afternoon observations, with daily global coverage, a
spectral resolution and signal-to-noise ratio (SNR) equivalent to OMI, but combined with a spatial resolution
improved by an order of magnitude, which potentially offers an unprecedented view of the spatiotemporal
variability of NMVOC emissions.
To fully exploit the potential of satellite data, applications relying on tropospheric HCHO observations require
high quality long-term time series, provided with well characterized errors and averaging kernels, and
consistently retrieved from the different sensors. Furthermore, as the HCHO observations are aimed to be used
synergistically with other species observations (e.g. with $NO_2$ for air quality applications), it is essential to
homogenize as much as possible the retrieval methods as well as the external databases, in order to minimize
systematic biases between the observations. The design of the TROPOMI HCHO prototype algorithm,
developed at BIRA-IASB, has been driven by the experience developed with formaldehyde retrievals from the
series of precursor missions OMI, GOME(-2) and SCIAMACHY. Furthermore, within the S5P Level 2
Working Group project (L2WG), a strong component of verification has been developed involving independent
retrieval algorithms for each operational prototype algorithm. For HCHO, the University of Bremen (IUP-UB)
has been responsible of the algorithm verification. An extensive comparison of the processing chains of the
prototype (the retrieval algorithm presented in this paper) and verification algorithm has been conducted. In
parallel, within the EU FP7-project Quality Assurance for Essential Climate Variables (QA4ECV, Lorente et
al., 2017), a detailed step by step study has been performed for HCHO and $NO_2$ DOAS retrievals, including
more scientific algorithms (BIRA-IASB, IUP-UB, MPIC, KNMI and WUR), leading to state-of-the art
European products (www.qa4ecv.eu). Those iterative processes led to improvements that have been included
in the S5P prototype algorithm, or are proposed as options for future improvements of the operational
algorithm.





This paper gives a thorough description of the TROPOMI HCHO algorithm baseline, as implemented at the
German Aerospace Center (DLR) in the S5P operational processor UPAS-2 (Universal Processor for UV/VIS
Atmospheric Spectrometers). It reflects the S5P HCHO Level 2 Algorithm Theoretical Basis Document v1.0
and also describes the options to be activated after the S5P launch, as implemented for the QA4ECV OMI
HCHO retrieval algorithm (see illustration in Figure 1).
In Section 2, we discuss the product requirements and the expected product performance in terms of precision
and trueness, and provide a complete description of the retrieval algorithm. In Section 3, the uncertainty of the
retrieved columns and the error budget is presented. Results from the algorithm verification exercise are given
in Section 4. The possibilities and needs for future validation of the retrieved HCHO data product can be found
in Section 5. Conclusions are given in Section 6.

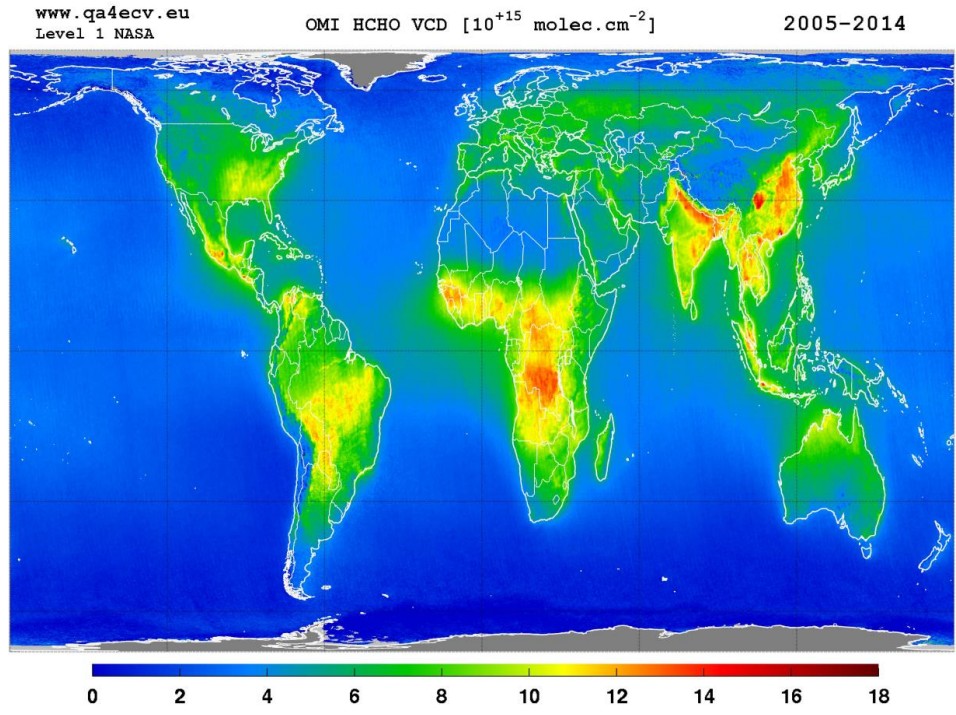


**Figure 1: 10-years average of HCHO vertical columns retrieved from OMI between 2005 and 2014**
**(http://www.qa4ecv.eu/ecv/hcho-p/data).**



## 2. TROPOMI HCHO algorithm

### 2.1 Product Requirements

In the UV, the sensitivity to HCHO concentrations in the boundary layer is intrinsically limited from space due to the combined effect of Rayleigh and Mie scattering that limit the fraction of radiation scattered back from low altitudes and reflected from the surface to the satellite. In addition, ozone absorption reduces the number of photons that reach the lowest atmospheric layers. Furthermore, the absorption signatures of HCHO are weaker than those of other UV-Vis absorbers, such as e.g. $NO_2$. As a result, the retrieval of formaldehyde from space is noise sensitive and error prone. While the precision (or random uncertainty) is mainly driven by the signal to noise ratio of the recorded spectra, the trueness (or systematic uncertainty) is limited by the current knowledge on the external parameters needed in the different retrieval steps.

The requirements for HCHO retrievals have been identified as part of the TROPOMI science objectives document (van Weele et al., 2008), the COPERNICUS Sentinels-4/-5 Mission Requirements Document MRD (Langen et al., 2011; 2017), and the S5P Mission Advisory Group report of the review of user requirements for Sentinels-4/-5 (Bovensmann et al., 2011). The requirements for HCHO are summarised in Table 1. Uncertainty requirements include retrieval errors as well as measurement (instrument-related) errors. Absolute requirements (in total column units) relate to background conditions, while percentage values relate to elevated columns.

Three main COPERNICUS environmental themes have been defined as ozone layer (A), air quality (B), and climate (C) with further division into sub themes. Requirements for HCHO have been specified for a number of these sub themes (B1: Air Quality Protocol Monitoring, B2: Air Quality Near-Real Time, B3: Air Quality Assessment, and C3: Climate Assessment). With respect to air quality protocol monitoring, which is mostly concerned with trend and variability analysis, the requirements are specified for NMVOC emissions on monthly to annual time scales and for larger region/country scale (Bovensmann et al., 2011). In the error analysis section, we discuss these requirements and the expected performances of the HCHO retrieval algorithm.

**Table 1: Requirements on HCHO vertical tropospheric column products as derived from the MRD. Where numbers are given as "a - b", the first is the target requirement and the second is the threshold requirement.**

| Horizontal resolution [km] | Revisit time | Theme | Required uncertainty |
|---|---|---|---|
| 5-20 | 0.5-2h | B1, B2, B3 | 30-60% or $1.3 \times 10^{15}$ molec.cm$^{-2}$ (least stringent) |
| 5-50 | 6 - 24x3 hour | C3 | 30% or $1.3 \times 10^{15}$ molec.cm$^{-2}$ (least stringent) |





## 117    2.2 Algorithm description

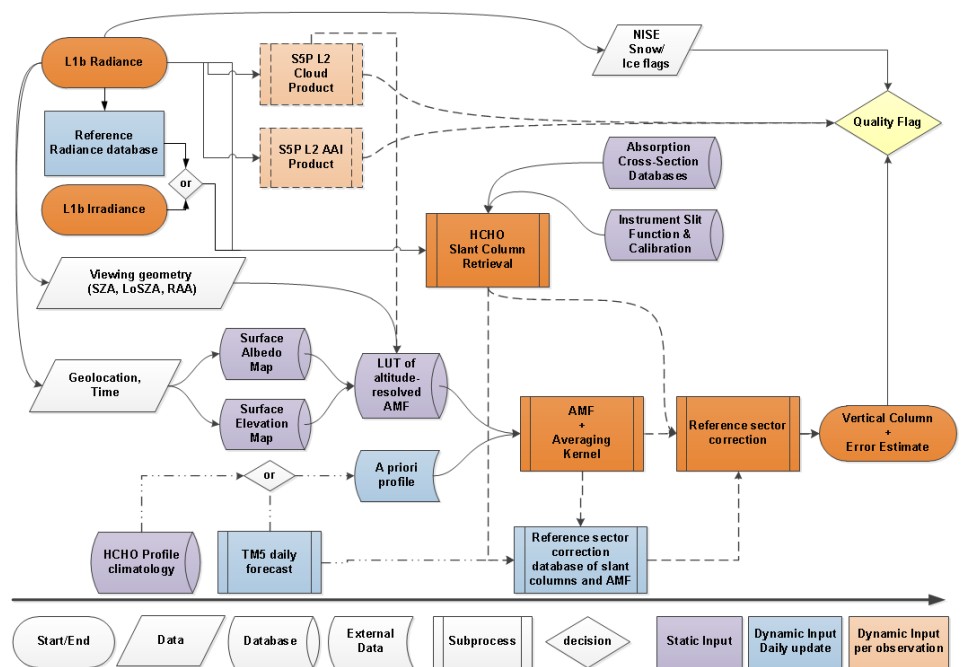

**Figure 2: Flow Diagram of the L2 HCHO retrieval algorithm implemented in the S5P operational processor.**

Figure 2 displays a flow diagram of the level-2 (L2) HCHO retrieval algorithm implemented in the S5P
operational processor. The baseline operation flow scheme is based on the Differential Optical Absorption
Spectroscopy (DOAS) retrieval method (Platt et al., 1994; Platt and Stutz, 2008; and references therein). It is
identical in concept to the one of $SO_2$ (Theys et al., 2017) and very close to the one of $NO_2$ (van Geffen et al.,
2017). The interdependencies with auxiliary data and other L2 retrievals, such as clouds, aerosols or surface
reflectance are also represented.
Following the diagram in Figure 2, the processing of S5P level-1b (L1b) data proceeds as follows: radiance
and irradiance spectra are read from the L1b file, along with geolocation data such as pixel coordinates and
observation geometry (sun and viewing angles). The relevant absorption cross section data as well as
characteristics of the instrument are used as input for the determination of the HCHO slant columns ($N_s$). In
parallel to the slant column fit, S5P cloud information and absorbing aerosol index (AAI) data are obtained
from the operational chain. Alongside, in order to convert the slant column to a vertical column ($N_v$), an air
mass factor ($M$) that accounts for the average light path through the atmosphere is calculated. For this purpose,
several auxiliary data are read from external (operational and static) sources: cloud cover data, topographic
information, surface albedo, and the a priori shape of the vertical HCHO profile in the atmosphere. The AMF
is computed by combining an a priori formaldehyde vertical profile and altitude-resolved air mass factors
extracted from a pre-computed look-up-table (also used as a basis for the error calculation and retrieval
characterization module). This look up table has been created using the VLIDORT 2.6 radiative transfer model



(Spurr et al., 2008a) at a single wavelength representative for the retrieval interval. It is used to compute the
total column averaging kernels (Eskes and Boersma, 2003), which provide essential information on the
measurement vertical sensitivity and are required for comparison with other types of data.
Background normalization of the slant columns is required in the case of weak absorbers such as formaldehyde.
Before converting the slant columns into vertical columns, background values of $N_s$ are normalized to
compensate for possible systematic offsets (reference sector correction, see below). The tropospheric vertical
column end product results therefore from a differential column to which is added the HCHO background due
to methane oxidation, estimated using a tropospheric chemistry transport model.
The final tropospheric HCHO vertical column is obtained using the following equation:

$$N_v = \frac{N_s - N_{s,0}}{M} + N_{v,0}$$

(1)

The main outputs of the algorithm are the slant column density ($N_s$), the tropospheric vertical column ($N_v$), the
tropospheric air mass factor ($M$), and the values used for the reference sector correction ($N_{s,0}$ and $N_{v,0}$).
Complementary product information includes the clear sky air mass factor, the error on the total column, the
averaging kernel, and quality flags. Table 13 in the appendix B gives a non-exhaustive set of data fields that
are provided in the level 2 data product. A complete description of the level 2 data format is given in the S5P
HCHO Product User Manual (Pedergnana et al., 2017).
Algorithmic steps are described in more details in the next sections, and settings are summarized in Table 2,
along with algorithmic improvements developed in the framework of the EU FP7-project QA4ECV and
proposed for future TROPOMI processor updates. Figure 3 also presents examples of monthly averaged
HCHO vertical columns over four NMVOC emission regions, along with the background correction values.
**Table 2 : Summary of algorithm settings used to retrieve HCHO tropospheric columns from**
**TROPOMI spectra. The last column lists additional features implemented in the QA4ECV HCHO**
**product, which are options for future updates of the S5P Processor.**

| Parameter | S5P Operational Algorithm | QA4ECV Algorithm |
|---|---|---|
| **Slant Columns** | | |
| **Fitting interval-1** | 328.5-359 nm | |
| **Fitting interval-2** | 328.5-346 nm ($N_{s, BrO}$ fixed by fit in interval-1) | |
| **Absorption cross-sections** | HCHO, Meller and Moortgat (2000), 298K <br> NO₂, Vandaele et al. (1998), 220K <br> Ozone, Serdyuchenko et al. (2013), 223 + 243K <br> BrO, Fleischmann et al. (2004), 223K <br> O₂-O₂, Thalman et al. (2013), 293K | |
| **Ring effect** | Ring cross-section based on the technique outlined by Chance et al. (1997), defined as $I_{rrs}/I_{elas}$, where $I_{rrs}$ and $I_{elas}$ are the intensities for inelastic (Rotational Raman Scattering; RRS) and elastic scattering processes. | |
| **Non-linear O3 absorption effect** | 2 pseudo-cross sections from the Taylor expansion of the ozone slant column into wavelength and the O₃ vertical optical depth (Puķīte et al., 2010). | |



| Slit function | One slit function per binned spectrum as a function of wavelength (Pre Flight Model, TROPOMI ISRF Calibration Key Data v1.0.0) | Fit of a prescribed function shape to determine the ISRF during wavelength calibration + online convolution of cross-sections. |
|---|---|---|
| Polynomial | 5th order | |
| Intensity offset correction | Linear offset $(1/I_0)$ | |
| Iterative spike removal | Not activated. | Activated. Tolerance factor 5 (see section 2.2.1) |
| Reference spectrum $I_0$ | Daily solar irradiance | Daily average of radiances, per row, selected in a remote region. |
| **Air Mass Factors** | | |
| Altitude dependent AMFs | VLIDORT , 340 nm, 6-D AMF look-up table | |
| Treatment of partly cloudy scenes | IPA, no correction for $f_{eff}$ <10% | |
| Aerosols | No explicit correction | |
| A priori profile shapes | TM5-MP 1°x1°, daily forecast (NRT) or reprocessed (Offline) | |
| Correction of surface pressure | Yes (Equation (10)) | |
| Surface Albedo | OMI-based monthly minimum LER (update of Kleipool et al., 2008) | |
| Digital elevation map | GMTED2010 (Danielson et al., 2011) | |
| Cloud product | S5P operational cloud product, treating clouds as Lambertian reflectors (OCRA/ROCINN-CRB, Loyola et al., 2017) | OMI operational cloud algorithm, treating clouds as Lambertian reflectors ($O_2$-$O_2$ , Veefkind et al., 2016) |
| **Background Correction** | | |
| Correction equation | $N_{v,0} = N_{v,0,CTM}$ | $N_{v,0} = \frac{M_0}{M} N_{v,0,CTM}$ (see section 2.2.3) |




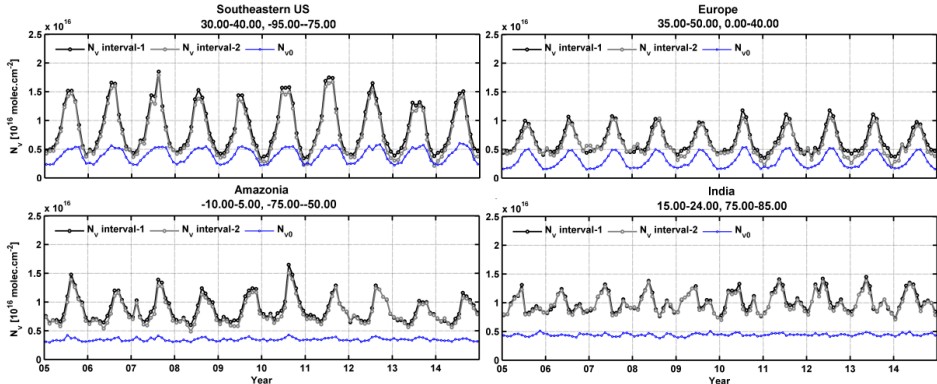


**Figure 3: Example of regional and monthly averages of the HCHO vertical columns over different**
**NMVOC emission regions, for the period 2005-2014. Results of the retrievals in the two fitting**
**intervals (-1 and -2) are shown, as well as the magnitude of the background vertical column ($N_{v,0}$).**
**2.2.1 Formaldehyde slant column retrieval**
The DOAS method relies on the application of Beer-Lambert's law. The backscattered earthshine spectrum as
measured by the satellite spectrometer contains the strong solar Fraunhofer lines and additional fainter features
due to interactions taking place in the Earth atmosphere during the incoming and outgoing paths of the
radiation. The basic idea of the DOAS method is to separate broad and narrowband spectral structures of the
absorption spectra in order to isolate the narrow trace gas absorption features. In practice, the application of
the DOAS approach to scattered light observations relies on the following key approximations:
1.     For weak absorbers the exponential function can be linearized and the Lambert-Beer law can be

applied to the measured radiance to which a large variety of atmospheric light paths contributes.

2.     The absorption cross-sections are assumed to be weakly dependent on temperature and

independent of pressure. This allows expressing light attenuation in terms of Beer-Lambert's law,

and (together with approximation 1) separating spectroscopic retrievals from radiative transfer

calculations by introducing the concept of one effective slant column density for the considered

wavelength window.

3.     Broadband variations are approximated by a common low-order polynomial to compensate for

the effects of loss and gain from scattering and reflections by clouds/air molecules and/or at the

Earth surface.

The DOAS equation is obtained by considering the logarithm of the radiance $I(\lambda)$ and the irradiance $E_0(\lambda)$ (or
another reference radiance selected in a remote sector) and including all broadband variations in a polynomial
function:

$$\ln\frac{I(\lambda)}{E_0(\lambda)} \cong -\sum_j \sigma_j(\lambda)\, N_{s,j} + \sum_p c_p \lambda^p \tag{2}$$



$$\tau_s^{meas}(\lambda) \cong \tau_s^{diff}(\lambda, N_{s,j}) + \tau_s^{smooth}(\lambda, c_p), \tag{3}$$

where the measured optical depth $\tau_s^{meas}$ is modelled using a highly structured part $\tau_s^{diff}$ and a broadband
variation $\tau_s^{smooth}$.
Equation (2) is a linear equation between the logarithm of the measured quantities ($I$ and $E_0$), the slant column
densities of all relevant absorbers ($N_{s,j}$) and the polynomial coefficients ($c_p$), at multiple wavelengths. DOAS
retrievals consist in solving an over-determined set of linear equations, which can be done by standard methods
of linear least squares fit (Platt and Stutz, 2008). The fitting process consists in minimizing the chi-square
function, i.e. the weighted sum of squares derived from Equation (3):

$$X^2 = \sum_{i=1}^{k} \frac{\left(\tau_s^{meas}(\lambda_i) - \tau_s^{diff}(\lambda_i, N_{s,j}) - \tau_s^{smooth}(\lambda_i, c_p)\right)^2}{\varepsilon_i^2} \tag{4}$$

where the summation is made over the individual spectral pixels included in the selected wavelength range (k
is the number of spectral pixels in the fitting interval). $\varepsilon_i$ is the statistical uncertainty on the measurement at
wavelength $\lambda_i$. Weighting the residuals by the instrumental errors $\varepsilon_i$ is optional. When no measurement
uncertainties are used (or no error estimates are available), all uncertainties in Equation (4) are set to $\varepsilon_i = 1$,
giving all measurement points equal weight in the fit.
In order to optimize the fitting procedure, additional structured spectral effects have to be considered carefully
such as the Ring effect (Grainger and Ring, 1962). Furthermore, the linearity of Equation (3) may be broken
down by instrumental aspects such as small wavelength shifts between I and $E_0$.
**Fitting intervals, absorption cross-sections and spectral fitting settings**
Despite the relatively large abundance of formaldehyde in the atmosphere (of the order of $10^{16}$ molec.cm$^{-2}$)
and its well-defined absorption bands, the fitting of HCHO slant columns in earthshine radiances is a challenge
because of the low optical density of HCHO compared to other UV-Vis absorbers. The typical HCHO optical
density is one order of magnitude smaller than that of $NO_2$ and three orders of magnitude smaller than that for
$O_3$ (see Figure 4). Therefore, the detection of HCHO is limited by the signal to noise ratio of the measured
radiance spectra and by possible spectral interferences and misfits due to other molecules absorbing in the same
fitting interval, mainly ozone, BrO and $O_4$. In general, the correlation between cross-sections decreases if the
wavelength interval is extended, but the assumption of a single effective light path defined for the entire
wavelength interval may not be fully satisfied, leading to systematic misfit effects that may also introduce
biases in the retrieved slant columns. To optimize DOAS retrieval settings, a trade-off has to be found
minimising these effects taking also into consideration the instrumental characteristics. A basic limitation of
the classical DOAS technique is the assumption that the atmosphere is optically thin in the wavelength region
of interest. At shorter wavelengths, the usable spectral range of DOAS is limited by rapidly increasing Rayleigh
scattering and $O_3$ absorption. The DOAS assumptions start to fail for ozone slant columns larger than 1500 DU
(Van Roozendael et al., 2012). Historically, different wavelength intervals have been selected between 325 and
360 nm for the retrieval of HCHO using previous satellite UV spectrometers (e.g: GOME, Chance et al., 2000;
SCIAMACHY, Wittrock et al., 2006, or GOME-2, Vrekoussis et al., 2010). The TEMIS dataset combines
HCHO observations from GOME, SCIAMACHY, GOME-2 and OMI measurements retrieved in the same
interval (De Smedt et al., 2008; 2012; 2015). The NASA operational and PCA OMI algorithm exploit a larger
interval (Kurosu, 2008; González Abad et al., 2015, Li et al., 2015). The latest QA4ECV product uses the
largest interval, thanks to the good quality of the OMI level 1 spectra. A summary of the different wavelength
intervals is provided in Table 3.

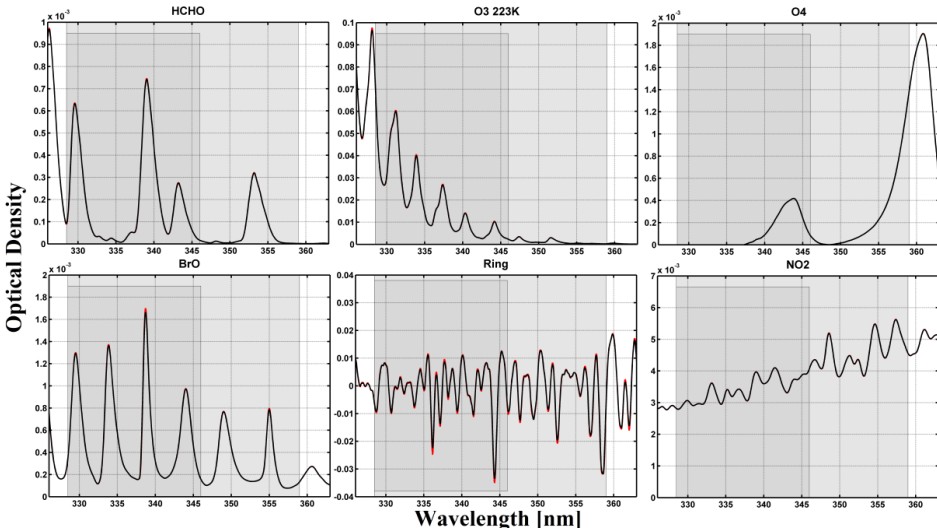


**Figure 4: Typical optical densities of HCHO, $O_3$, $O_2$-$O_2$, BrO, Ring effect, and $NO_2$ in the near UV. The slant columns have been taken as $1.3 \times 10^{16}$ molec.cm$^{-2}$ for HCHO, $10^{19}$ molec.cm$^{-2}$ for $O_3$, $0.4 \times 10^{43}$ molec$^2$.cm$^{-5}$ for $O_2$-$O_2$, $10^{14}$ molec.cm$^{-2}$ for BrO, and $1 \times 10^{16}$ molec.cm$^{-2}$ for $NO_2$. High resolution absorption cross-sections of Table 2 have been convolved with the TROPOMI ISFRs v1.0 (row 1 is shown in red and row 225 in black, see also Figure 5). The two fitting intervals (-1 and -2) used to retrieve HCHO slant columns are limited by grey areas.**

**Table 3: Wavelength intervals used in previous formaldehyde retrieval studies [nm].**

|  | GOME | SCIAMACHY | GOME-2 | OMI |
|---|---|---|---|---|
| **Chance et al., 2000** | 337.5-359 |  |  |  |
| **Wittrock et al., 2006** |  | 334-348 |  |  |
| **Vrekoussis et al., 2010** |  |  | 337-353 |  |
| **Hewson et al., 2012** |  |  | 328.5-346 |  |
| **González Abad et al., 2015; Li et al., 2015** |  |  |  | 328.5-356.5 |
| **De Smedt et al., 2008 ; 2012 ; 2015** | 328.5-346 | 328.5-346 | 328.5-346 (BrO in 328.5-359) | 328.5-346 (BrO in 328.5-359) |
| **QA4ECV** |  |  |  | 328.5-359 |



As for the TEMIS OMI HCHO product (De Smedt et al., 2015), the TROPOMI L2 HCHO retrieval algorithm

includes a two-step DOAS retrieval approach, based on two wavelength intervals:

1.    328.5-359 nm: This interval includes six BrO absorption bands and minimizes the correlation with HCHO, allowing a significant reduction of the retrieved slant column noise. Note that this interval includes part of a strong $O_4$ absorption band around 360 nm, which may introduce geophysical artefacts of HCHO columns over arid soils or high altitude regions.

2.    328.5-346 nm: in a second step, HCHO columns are retrieved in a shorter interval, but using the BrO slant column values determined in the first step. This approach allows to efficiently de-correlate BrO from HCHO absorption while, at the same time, the $O_4$-related bias is avoided.

The use of a large fitting interval generally allows for a reduction of the noise on the retrieved slant columns.
However, a substantial gain can only be obtained if the level 1 spectra are of sufficiently homogeneous quality
over the full spectral range. Indeed, experience with past sensors not equipped with polarization scramblers
(e.g. GOME(-2) or SCIAMACHY) has shown that this gain can be partly or totally overruled due to the impact
of interfering spectral polarization structures (De Smedt et al., 2012; 2015). Assuming spectra free of spectral
features, the QA4ECV baseline option using one single large interval (fitting interval-1) will be applicable to
TROPOMI. Results of the retrievals from the two intervals applied to OMI are presented in Figure 3. In this
case, vertical column differences between the two intervals are generally lower than 10%. They can however
reach 20% in winter time.
In both intervals, the absorption cross-sections of $O_3$ at 223K and 243K, $NO_2$, BrO and $O_4$ are included in the
fit. The correction for the Ring effect, defined as $I_{rrs}/I_{elas}$, where $I_{rrs}$ and $I_{elas}$ are the intensities for inelastic
(Rotational Raman Scattering; RRS) and elastic scattering processes, is based on the technique published by
Chance et al. (1997). Furthermore, in order to better cope with the strong ozone absorption at wavelengths
shorter than 336 nm, the method of Puķīte et al. (2010) is implemented. In this method, the variation of the
ozone slant column over the fitting window is taken into account. At the first order, the method consists in
adding two cross-sections to the fit: $\lambda\sigma_{\_O3}$ and $\sigma_{\_O3}^2$ (Puķīte et al., 2010; De Smedt et al.; 2012), using the $O_3$
cross-sections at 223K (close to the temperature at ozone maximum in the tropics). It allows a much better
treatment of optically thick ozone absorption in the retrieval and therefore to reduce the systematic
underestimation of the HCHO slant columns by 50 to 80%, for SZA from 50° to 70°.
To obtain the optical density (Equation (2)), the baseline option is to use the daily solar irradiance. A more
advanced option, implemented in QA4ECV, is to use daily averaged radiances, selected for each detector row,
in the equatorial Pacific (Lat: [-5° 5°], Long: [180° 240°]). The main advantages of this approach are (1) an
important reduction of the fit residuals (by up to 40%) mainly due to the cancellation of $O_3$ absorption and
Ring effect present in both spectra; (2) the fitted slant columns are directly corrected for background offsets
present in both spectra; (3) possible row-dependent biases (stripes) are directly corrected owing to the use of
one reference per detector row; and (4) the sensitivity to instrument degradation is reduced because degradation
effects tend to cancel between the analyzed spectra and the references that are used. It must be noted however
that the last three effects can be mitigated when a solar irradiance is used as reference, by means of a post-



processing treatment applied as part of the background correction of the slant columns (see section 2.2.3). The
option of using an equatorial radiance as reference will be activated in the operational processor after the launch
of TROPOMI, during the commissioning phase of the instrument.
**Wavelength calibration and convolution to TROPOMI resolution**
The quality of the DOAS fit critically depends on the accuracy of the wavelength alignment between the
earthshine radiance spectrum, the reference (solar irradiance) spectrum and the absorption cross sections. The
wavelength registration of the reference spectrum can be fine-tuned to an accuracy of a few hundredths of a
nanometer by means of a calibration procedure making use of the solar Fraunhofer lines. To this end, a
reference solar atlas $E_s$ accurate in wavelength to better than 0.01 nm (Chance and Kurucz, 2010) is degraded
to the resolution of the instrument, through convolution by the TROPOMI instrumental slit function (see Figure

5).

Using a non-linear least-squares approach, the shift ($\Delta_i$) between the TROPOMI irradiance and the reference
solar atlas is determined in a set of equally spaced sub-intervals covering a spectral range large enough to
encompass all relevant fitting intervals. The shift is derived according to the following equation:

$$E_0(\lambda) = E_s(\lambda - \Delta_i)$$

(5)

where $E_s$ is the reference solar spectrum convolved at the resolution of the TROPOMI instrument and $\Delta_i$ is the
shift in sub-interval $i$. A polynomial is fitted through the individual points to reconstruct an accurate wavelength
calibration $\Delta(\lambda)$ over the complete analysis interval. Note that this approach allows compensating for stretch
and shift errors in the original wavelength assignment. In the case of TROPOMI (or OMI), the procedure is
complicated by the fact that such calibrations must be performed and stored for each separate spectral field on
the CCD detector array. Indeed due to the imperfect characteristics of the imaging optics, each row of the
instrument must be considered as a separate detector for analysis purposes.
In a subsequent step of the processing, the absorption cross-sections of the different trace gases must be
convolved with the instrumental slit functions. The baseline approach is to use slit functions determined as part
of the TROPOMI key data. Slit functions, or Instrument Spectral Response Functions (ISRF), are delivered for
each binned spectrum and as a function of the wavelength as illustrated in Figure 5. Note that an additional
feature of the prototype algorithm allows to dynamically fit for an effective slit function of known line shape.
This can be used for verification and monitoring purpose during commissioning and later on during the mission.
This option is used for the QA4ECV OMI HCHO product.
More specifically, wavelength calibrations are made for each orbit as follows:
• The irradiances (one for each binned row of the CCD) are calibrated in wavelength over the 325-360
nm wavelength range, using 5 sub-windows.



• The earthshine radiances are first interpolated on the original L1 irradiance grid. The irradiance
calibrated wavelength grid is assigned to those interpolated radiance values.
• The absorption cross-sections are interpolated (cubic spline interpolation) on the calibrated
wavelength grid, prior to the analysis.
• In the case where averaged radiances are used as reference, an additional step must be performed: the
cross-sections are aligned to the reference spectrum by means of shift/stretch values derived from a
least-squares fit of the calibrated irradiance towards the averaged reference radiance.
• During spectral fitting, shift and stretch parameters for the radiance are derived, to align each radiance
with cross sections and reference spectrum.

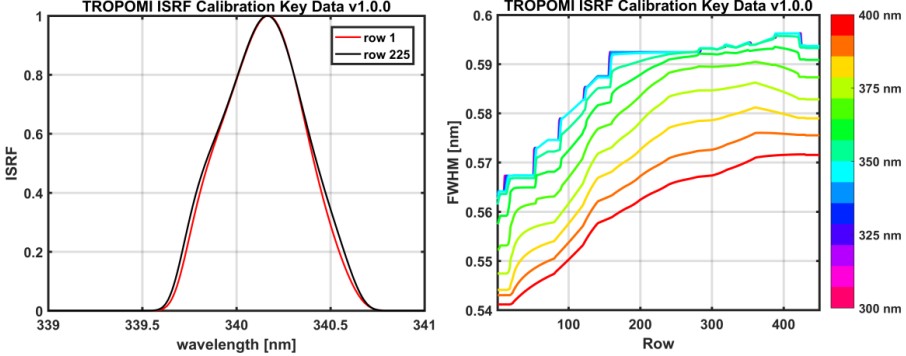


**Figure 5: Right panel: Examples of TROPOMI slit functions around 340 nm, for row 1 and row 225.**
**Left panel: TROPOMI spectral resolution in channel 3, as a function of the row and the wavelength,**
**derived from the instrument key data ISFR v1.0.0.**


**Spike removal algorithm**

A method to remove individual hot pixels or pixels affected by the South Atlantic Anomaly has been presented
for NO$_2$ retrievals in Richter et al. (2011). Often only a few individual detector pixels are affected and in these
cases, it is possible to identify and remove the outliers from the fit. However, as the amplitude of the distortion
is usually only of the order of a few percent or less, it cannot always be found in the highly structured spectra
themselves. Higher sensitivity for spikes can be achieved by analysing the residual of the fit where the
contribution of the Fraunhofer lines, scattering, and absorption is already removed. When the residual for a
single pixel exceeds the average residual of all pixels by a chosen threshold ratio (the tolerance factor), the
pixel is excluded from the analysis, in an iterative process. This procedure is repeated until no further outliers
are identified, or until the maximum number of iterations is reached (here fixed to 3). Tests performed with
OMI spectra show that a tolerance factor of 5 improves the HCHO fits. This is especially important to handle
the sensitivity of 2-D detector arrays to high energy particles. However, this improvement of the algorithm has
a non-negligible impact on the time of processing (x 1.8). This option is activated in the QA4ECV algorithm,
and will be activated in the TROPOMI operational algorithm in the next update of the processor.



**2.2.2 Tropospheric air mass factor**
In the DOAS approach, an optically thin atmosphere is assumed. The mean optical path of scattered photons
can therefore be considered as independent of the wavelength within the relatively small spectral interval
selected for the fit. One can therefore define a single effective air mass factor given by the ratio of the slant to
the vertical optical depth of a particular absorber $j$:

$$M_j = \frac{\tau_{s,j}}{\tau_{v,j}} . \tag{6}$$

In the troposphere, scattering by air molecules, clouds and aerosols leads to complex light paths and therefore
complex altitude-dependent air mass factors. Full multiple scattering calculations are required for the
determination of the air mass factors, and the vertical distribution of the absorber has to be assumed *a priori*.
For optically thin absorbers, the formulation of Palmer et al. (2001) is conveniently used. It decouples the
height-dependent measurement sensitivity from the vertical profile shape of the species of interest, so that the
tropospheric AMF ($M$) can be expressed as the average of the altitude dependent air mass factors ($m_l$) weighted
by the partial columns ($n_{al}$) of the a priori vertical profile in each vertical layer $l$, from the surface up to the
tropopause index ($lt$):

$$M = \frac{\sum_{l=1}^{l=lt} m_l(\lambda, \theta_0, \theta, \varphi, A_s, p_s, f_c, A_{cloud}, p_{cloud}) n_{al}(lat, long, time)}{\sum_{l=1}^{l=lt} n_{al}(lat, long, time)}, \tag{7}$$

where $A_s$ is the surface albedo, $p_s$ is the surface pressure, and $f_c$, $A_{cloud}$ and $p_{cloud}$ are respectively the cloud
fraction, cloud albedo and cloud top pressure.
The altitude dependent air mass factors represent the sensitivity of the slant column to a change of the partial
columns $N_{v,j}$ at a certain level. In a scattering atmosphere, $m_l$ depends on the wavelength, the viewing angles,
the surface albedo, and the surface pressure, but not on the partial column amounts or the vertical distribution
of the considered absorber (optically thin approximation).
**LUT of altitude dependent air mass factors**
Generally speaking, $m$ depends on the wavelength, as scattering and absorption processes vary with
wavelength. However, in the case of HCHO, the amplitude of the $M$ variation is found to be small (less than
5% for SZA lower than 70°) in the 328.5-346 nm fitting window and a single air mass factor representative for
the entire wavelength interval is used at 340 nm (Lorente et al., 2017).
Figure 6 illustrates the dependency of $m$ with the observation angles, *i.e.* $\theta_0$ (a), $\theta$ (b), and $\varphi$ (c), and with scene
conditions like $A_s$ (d) and $p_s$ for a weakly (e) or highly reflecting surface (f). The decrease of sensitivity in the
boundary layer is more important for large solar zenith angles and wide instrumental viewing zenith angles.
The relative azimuth angle does have relatively less impact on the measurement sensitivity (note however that
aerosols and BRDF effects are not included in those simulations). In the UV, surfaces not covered with snow
have an albedo lower than 0.1, while snow and clouds generally present larger albedos. For a weakly reflecting





surface, the sensitivity decreases near the ground because photons are mainly scattered, and scattering can take
place at varying altitudes. Larger values of the surface albedo increase the fraction of reflected compared to
scattered photons, increasing measurement sensitivity to tropospheric absorbers near the surface. Over snow
or ice also multiple scattering can play an important role further increasing the sensitivity close to the surface.

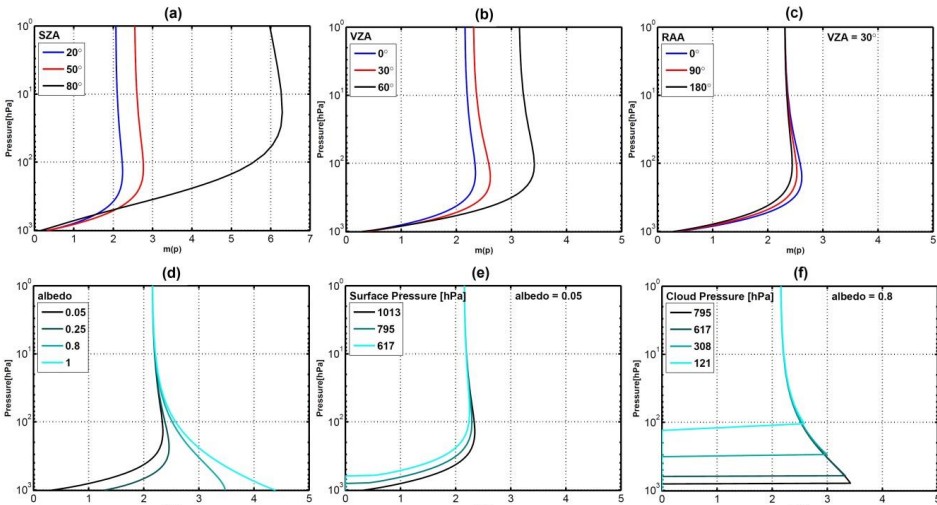


**Figure 6: Variation of the altitude dependent air mass factor with: (a) solar zenith angle, (b) viewing**
**zenith angle, (c) relative azimuth angle between the sun and the satellite, (d) surface albedo, (e) surface**
**pressure for a weakly reflecting surface, (f) surface pressure for a highly reflecting surface. Unless**
**specified, the parameters chosen for the radiative transfer simulations are: SZA=30°, VZA=0°,**
**RAA=0°, albedo=0.05, surface pressure=1063hPa, λ=340nm.**
Altitude dependent air mass factors are calculated with the VLIDORT v2.6 radiative transfer model (Spurr,
2008), at 340 nm, using an US standard atmosphere, for a number of representative viewing geometries, surface
albedos and surface pressures (used both for ground and cloud surface pressures), and stored in a look-up table.
Altitude dependent air mass factors are then interpolated within the lookup table for each particular observation
condition and interpolated vertically on the pressure grid of the a priori profile, defined within the TM5-MP
model (Williams et al., 2017). Linear interpolations are performed in $\cos(\theta_0)$, $\cos(\theta)$, relative azimuth angle
and surface albedo, while a nearest neighbour interpolation is performed in surface pressure. The parameter
values chosen for the look-up table are detailed in Table 4. In particular, the grid of surface pressure is very
thin near the ground, in order to minimise interpolation errors caused by the generally low albedo of ground
surfaces. Indeed, as illustrated by Figure 6 (e) and (f), the variation of the altitude dependent air mass factors
is more discontinuous with surface elevation (low reflectivity) than with cloud altitude (high reflectivity).
Furthermore, the LUT and model pressures are scaled to their respective surface pressures, in order to avoid
extrapolations outside the LUT range.





**Table 4: Parameters in the altitude dependent air mass factors lookup table**

| Parameter name | Nb. of grid points | Grid of values | Symbol |
|---|---|---|---|
| Solar zenith angle [°] | 17 | 0, 10, 20, 30, 40, 45, 50, 55, 60, 65, 70, 72, 74, 76, 78, 80, 85 | $\theta_0$ |
| Line of sight zenith angle [°] | 10 | 0, 10, 20, 30, 40, 50, 60, 65, 70, 75 | $\theta$ |
| Relative azimuth angle [°] | 5 | 0, 45, 90, 135, 180 | $\varphi$ |
| Surface albedo | 14 | 0, 0.01, 0.025, 0.05, 0.075, 0.1, 0.15, 0.2, 0.25, 0.3 0.4, 0.6, 0.8, 1.0 | $A_s$ |
| Surface pressure [hPa] | 17 | 1063.10, 1037.90, 1013.30, 989.28, 965.83, 920.58, 876.98, 834.99, 795.01, 701.21, 616.60, 540.48, 411.05, 308.00, 226.99, 165.79, 121.11 | $p_s$ |
| Atmospheric pressure [hPa] | 64 | 1056.77, 1044.17, 1031.72, 1019.41, 1007.26, 995.25, 983.38, 971.66, 960.07, 948.62, 937.31, 926.14, 915.09, 904.18, 887.87, 866.35, 845.39, 824.87, 804.88, 785.15, 765.68, 746.70, 728.18, 710.12, 692.31, 674.73, 657.60, 640.90, 624.63, 608.58, 592.75, 577.34, 562.32, 547.70, 522.83, 488.67, 456.36, 425.80, 396.93, 369.66, 343.94, 319.68, 296.84, 275.34, 245.99, 210.49, 179.89, 153.74, 131.40, 104.80, 76.59, 55.98, 40.98, 30.08, 18.73, 8.86, 4.31, 2.18, 1.14, 0.51, 0.14, 0.03, 0.01, 0.001 | $p_l$ |
| Altitude corresponding to the atmospheric pressure, using an US standard atmosphere [km] (for information) | 64 | -0.35, -0.25, -0.15, -0.05, 0.05, 0.15, 0.25, 0.35, 0.45, 0.55, 0.65, 0.75, 0.85, 0.95, 1.10, 1.30, 1.50, 1.70, 1.90, 2.10, 2.30, 2.50, 2.70, 2.90, 3.10, 3.30, 3.50, 3.70, 3.90, 4.10, 4.30, 4.50, 4.70, 4.90, 5.25, 5.75, 6.25, 6.75, 7.25, 7.75, 8.25, 8.75, 9.25, 9.75, 10.50, 11.50, 12.50, 13.50, 14.50, 16.00, 18.00, 20.00, 22.00, 24.00, 27.50, 32.50, 37.50, 42.50, 47.50, 55.00, 65.00, 75.00, 85.00, 95.00 | $z_l$ |

**Treatment of partly cloudy scenes**
The AMF calculations for TROPOMI will use the cloud fraction ($f_c$), cloud albedo ($A_{cloud}$) and cloud pressure
($p_{cloud}$) from the S5P operational cloud retrieval, treating clouds as Lambertian reflectors (OCRA/ROCINN-
CRB, Loyola et al., 2017). The applied cloud correction is based on the independent pixel approximation
(Martin et al., 2002 and Boersma et al., 2004), in which a inhomogeneous satellite pixel is considered as a
linear combination of two independent homogeneous scenes, one completely clear and the other completely
cloudy. The intensity measured by the instrument for the entire scene is decomposed into the contributions
from the clear-sky and cloudy fractions. Accordingly, for each vertical layer, the altitude dependent air mass





factor of a partly cloudy scene is a combination of two air mass factors, calculated respectively for the cloud-
free and cloudy fractions of the scene:

$$m_l = (1 - w_c)m_{l\_clear}(A_s, p_s) + w_c m_{l\_cloud}(A_{cloud}, p_{cloud}) \tag{8}$$

where $m_{l\_clear}$ is the altitude dependent air mass factor for a completely cloud-free pixel, $m_{l\_cloud}$ is the altitude
dependent air mass factor for a completely cloudy scene, and the cloud radiance fraction $w_c$ is defined as:

$$w_c = \frac{f_c I_{cloud}(A_{cloud}, p_{cloud})}{(1 - f_c)I_{clear}(A_s, p_s) + f_c I_{cloud}(A_{cloud}, p_{cloud})} \tag{9}$$

$I_{clear}$ and $I_{cloud}$ are respectively the radiance intensities for clear-sky and cloudy scenes whose values are
calculated with VLIDORT at 340 nm and stored in look-up tables with the same grids as the altitude dependent
air mass factors. $m_{l\_clear}$ and $I_{clear}$ are evaluated for a surface albedo $A_s$ and a surface pressure $p_s$, while
$m_{l\_cloud}$ and $I_{cloud}$ are estimated for a cloud albedo $A_{cloud}$ and at the cloud pressure $p_{cloud}$. Note that the
variations of the cloud albedo are directly related to the cloud optical thickness. Strictly speaking in a
Lambertian (reflective) cloud model approach, only thick clouds can be represented (one should keep in mind
that still the penetration of photons into the cloud is not covered by the Lambertian model). An effective cloud
fraction corresponding to an effective cloud albedo of 0.8 ($f_{eff} = f_c \frac{A_c}{0.8}$) can be defined, in order to transform
optically thin clouds into equivalent optically thick clouds of reduced horizontal extent. In such altitude
dependent air mass factor calculations, a single cloud top pressure is assumed within a given viewing scene.
For low effective cloud fractions ($f_{eff}$ lower than 10%), the cloud top pressure retrieval is generally highly
unstable and it is therefore reasonable to consider the observation as a clear-sky pixel (i.e. the cloud fraction is
set to 0) in order to avoid unnecessary error propagation through the retrievals. This 10% threshold might be
adjusted according to the quality of the cloud product (Veefkind et al., 2016; Loyola et al., 2017).
It should be noted that this formulation of the altitude dependent air mass factor for a partly cloudy pixel
implicitly includes a correction for the HCHO column lying below the cloud and therefore not seen by the
satellite, the so-called "ghost column". Indeed, the total AMF calculation as expressed by (7) and (8) assumes
the same a priori vertical profile in both cloudy and clear parts of the pixel and implies an integration of the
profile from the top of atmosphere to the ground, for each fraction of the scene. The ghost column information
is thus coming from the a priori profiles. For this reason, observations with cloud fractions $f_{eff}$ larger than
30% are assigned with a poor quality flag and have to be used with caution.



**Aerosols**

The presence of aerosol in the observed scene may affect the quality of the retrieval. No explicit treatment of aerosols (absorbing or not) is foreseen in the operational algorithm as there is no general and easy way to treat the aerosols effect on the retrieval. At computing time, the aerosol parameters (extinction profile, single scattering albedo, ...) are unknown. However, the information on the AAI (Stein Zweers et al., 2016) will be included in the L2 HCHO files as it gives information to the user on the presence of absorbing aerosols and the affected data should be used and interpreted with care.

**A priori vertical profile shapes**

Formaldehyde concentrations decrease with altitude as a result of the near-surface sources of short-lived NMVOC precursors, the temperature dependence of $CH_4$ oxidation, and the altitude dependence of photolysis. The profile shape varies according to local NMHC sources, boundary layer depth, photochemical activity, and other factors.

To resolve this variability in the TROPOMI near-real time HCHO product, daily forecasts calculated with the TM5-MP chemical transport model (Huijnen et al., 2010, Williams et al., 2017) will be used to specify the vertical profile shape of the HCHO distribution. TM5-MP will also provide a priori profile shapes for the $NO_2$, $SO_2$, and CO retrievals. For the QA4ECV OMI products, high-resolution TM5-MP model runs were performed for the period 2004-2016, and the model profiles from this run are used for both HCHO and $NO_2$ retrievals.

TM5-MP is operated with a spatial resolution of 1°x1° in latitude and longitude, and with 34 sigma pressure levels up to 0.1hPa in the vertical direction. TM5-MP uses 3-hourly meteorological fields from the European Centre for Medium Range Weather Forecast (ECMWF) operational model (ERA-Interim reanalysis data for reprocessing, and the operational archive for real time applications and forecasts). These fields include global distributions of wind, temperature, surface pressure, humidity, and (liquid and ice) water content, and precipitation.

For the calculation of the HCHO air mass factors, the profiles are linearly interpolated in space and time, at pixel centre and local overpass time, through a model time step of 30 minutes. To reduce the errors associated to topography and the lower spatial resolution of the model compared to the TROPOMI 3.5x7 km$^2$ spatial resolution, the a priori profiles need to be rescaled to effective surface elevation of the satellite pixel. Following Zhou et al. (2009) and Boersma et al (2011), the TM5-MP surface pressure is converted by applying the hypsometric equation and the assumption that the temperature changes linearly with height:

$$p_s = p_{s,TM5}\left(\frac{T_{TM5}}{(T_{TM5} + \Gamma(z_{TM5} - z_s))}\right)^{-\frac{g}{R\Gamma}} \tag{10}$$

Where $p_{s,TM5}$ and $T_{TM5}$ are the TM5-MP surface pressure and temperature, $\Gamma = 0.0065 Km^{-1}$ the lapse rate, $z_{TM5}$ the TM5-MP terrain height, and $z_s$ surface elevation for the satellite ground pixel from a digital elevation





map at high resolution. $R=287\ J\ kg^{-1}\ K^{-1}$ is the gas constant for dry air, and $g = 9.8 ms^{-2}$ the gravitational
acceleration.

The pressure levels for the a priori HCHO profiles are based on the improved surface pressure level $p_s$:
$p_l = a_l + b_l p_s$ , $a_l$ and $b_l$ being the constants that effectively define the vertical coordinate (Table 13).

Yearly averaged air mass factors obtained using prior information summarized in Table 5, in particular TM5-MP HCHO profiles, are presented in Figure 7, in order to give an overview of the tropospheric AMF values and their global regional variations.

**Table 5: Prior information datasets used in the air mass factor calculation in the S5P HCHO operational algorithm and in the QA4ECV OMI algorithm.**

| Prior information | Origin of data set | Resolution | Symbol |
|---|---|---|---|
| **Surface Albedo** | OMI-based monthly minimum LER (update of Kleipool et al., 2008) | <ul><li>month</li><li>0.5°x0.5° (lat x long)</li><li>342 nm</li></ul> | $A_s$ |
| **Digital elevation map** | GMTED2010 (Danielson et al., 2011) | Average over the ground pixel area. | $z_s$ |
| **Cloud fraction** | Operational cloud product based on a Lambertian cloud model (S5P: Loyola et al., 2017; OMI: Veefkind et al., 2016). | For each ground pixel. | $f_c$ |
| **Cloud pressure** | | | $p_{cloud}$ |
| **Cloud albedo** | | | $A_{cloud}$ |
| **A priori HCHO profiles** | Forecast (NRT) or reanalysis from TM5-MP CTM | <ul><li>Daily profiles at overpass time</li><li>1°x1° (lat x long)</li><li>34 sigma pressure levels up to 0.1hPa</li></ul> | $n_a$ |





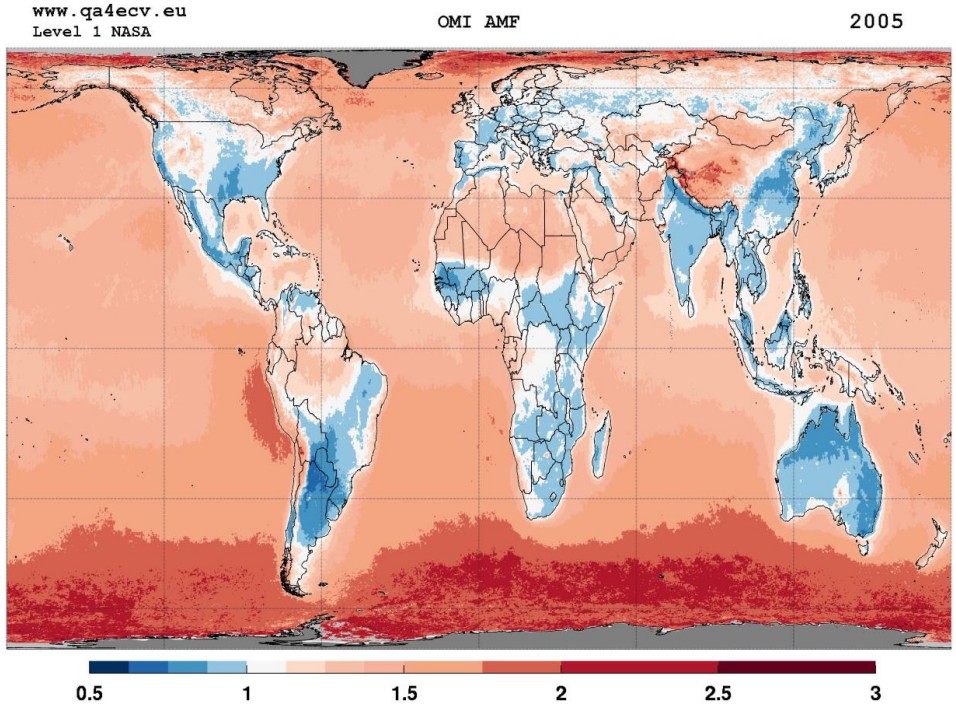

**Figure 7: Yearly averaged map of tropospheric air mass factors at 340 nm using the QA4ECV OMI HCHO algorithm. A priori HCHO profiles from high-resolution TM5-MP model runs have been used. The IPA cloud correction is applied for effective cloud fractions $f_{eff}$ larger than 10%. Observations with $f_{eff}$ larger than 30% have been filtered out.**

### 2.2.3 Across-track and zonal reference sector correction

Residual latitude-dependent biases in the columns, due to unresolved spectral interferences, are known to remain a limiting factor for the retrieval of weak absorbers such as HCHO. Retrieved HCHO slant columns can present large offsets depending on minor changes in the fit settings, and on minor instrumental spectral inaccuracies. Resulting offsets are generally global but also show particular dependencies, mainly with detector row (across-track) and with latitude (along-track). In the case of a 2D-detector array such as OMI or TROPOMI, across-track striping can possibly arise, due to imperfect calibration and different dead/hot pixel masks for the CCD detector regions. Offset corrections are also meant to handle some effects of the time-dependent degradation of the instrument.

A large part of the resulting systematic HCHO slant column uncertainty is reduced by the application of a background correction, which is based on the assumption that the background HCHO column observed over remote oceanic regions (Pacific Ocean) is only due to methane oxidation. The natural background level of HCHO is well estimated from chemistry model simulations of $CH_4$ oxidation ($N_{v,0,CTM}$). It is ranging from 2 to $4 \times 10^{15}$ molec.cm$^{-2}$, depending on the latitude and the season (De Smedt et al., 2008; 2015; González Abad et al., 2015).



474 For the HCHO retrieval algorithm, we use a 2-steps normalization of the slant columns (see Table 6):

475 • Across-track: the mean HCHO slant column is determined for each row in the reference sector around

476  the equator [-5° 5°], [180° 240°]. Data selection is based on the slant column errors from the DOAS

477  fit and on the cloud fraction (threshold values are given in Table 6). Those mean HCHO values are

478  subtracted from all the slant columns of the same day, as a function of the row. The aim is to reduce

479  possible row-dependent offsets. In the case were solar irradiance are used as reference, those offsets

480  can exceed $2\times10^{16}$ molec.cm$^{-2}$. They are reduced below $10^{15}$ molec.cm$^{-2}$ by this first step, or when

481  row averaged radiances are used as reference, as in the QA4ECV algorithm.

482 • Along-track: the latitudinal dependency of the across-track corrected HCHO SCs is modelled by a

483  polynomial fit through their mean values, all rows combined, in 5° latitude bins in the reference sector

484  ([-90° 90°], [180° 240°]). Again, data selection is based on the slant column errors from the DOAS

485  fit and on the cloud fraction.

486 These two corrections are applied to the global slant columns so that in the reference sector, the mean

487 background corrected slant columns ($\Delta N_s = N_s - N_{s,0}$) are centered around zero.

488 **Table 6: 2-steps normalization of the HCHO vertical columns**

| Correction | Region | Time frame | Column correction | Observation selection |
|---|---|---|---|---|
| **Across-track** | Equatorial Pacific Lat: [-5° 5°], Long: [180° 240°] | NRT: 1-week moving window | $dN_s(\text{row}) = N_s(\text{row}) - \overline{N_{s,0}}(\text{row})$ | $\sigma_{N_s} \leq 3\overline{\sigma_{N_s}}$ $f_c \leq 0.4$ |
| **Zonal Along-track** | Pacific Lat: [-90° 90°], Long: [180° 240°] | Offline: Daily correction | $\Delta N_s(lat) = dN_s(lat) - \overline{dN_{s,0}(lat)}$ $\overline{N_{s,0,CTM}}(lat) = \overline{M_0(lat)N_{v,0,CTM}(lat)}$ | $\overline{dN_{s,0}(lat)}$ $\leq 5e16$ |

489 To the corrected slant columns, the background HCHO values from a model have to be added. A latitude-

490 dependent polynomial is fitted daily through 5° latitude bin means of those modelled values in the reference

491 sector. Corresponding values are added to all the columns of the day. Strictly speaking, those background

492 values should be slant columns, derived as the product of air mass factors in the reference sector ($M_0$) with

493 HCHO vertical columns from the model ($N_{s,0,CTM} = M_0 N_{v,0,CTM}$) (González Abad et al., 2015). However, this

494 option requires the storage of the slant columns, the air mass factors, and their errors, in a separated database

495 (QA4ECV Algorithm and S5P option, see Equation (11)). An approximate solution is to add as background

496 the constant vertical column from the model ($N_{v,0,CTM}$), hence neglecting the variability of the $M_0/M$ ratio. This

497 is the current implementation in the S5P algorithm, which will be updated with equation (11) after launch. For

498 NRT purpose, the evaluation in the reference sector is made using a moving time window of 1 week. For offline

499 processing, the reference sector correction can be refined by using daily evaluations.

$$N_v = \frac{N_s - N_{s,0}}{M} + N_{v,0} = \frac{\Delta N_s}{M} + \frac{M_0}{M}N_{v,0,CTM} = \frac{\Delta N_s + N_{s,0,CTM}}{M} \quad\quad (11)$$





Figure 3 presents some examples of monthly and regionally averaged vertical columns, together with the
contribution of $N_{v,0}$. It should be realized that this contribution accounts for 20 to 50% of the vertical columns,
as expected from the large contribution of methane oxidation to the total HCHO column (Stavrakou et al.,
2015).

**3. Uncertainty analyses**
**3.1 Uncertainty formulation by uncertainty propagation**
The total uncertainty on the HCHO vertical column is composed of many sources of (random and systematic)
errors. In part those are related to the measuring instrument, such as errors due to noise or knowledge of the
slit function. In a DOAS-type algorithm, those instrumental errors propagate into the uncertainty of the slant
columns. Other types of error can be considered as model errors and are related to the representation of the
observation physical properties that are not measured. Examples of model errors are uncertainties on the trace
gas absorption cross-sections, the treatment of clouds and uncertainties of the a priori profiles. Model errors
can affect the slant columns, the air mass factors or the applied background corrections.
A formulation of the uncertainty can be derived analytically by error propagation, starting from the equation
of the vertical column (11) which directly results from the different retrieval steps. As the main algorithm steps
are performed independently, they are assumed to be uncorrelated. The total uncertainty on the tropospheric
vertical column can be expressed as (Boersma et al., 2004, De Smedt et al., 2008):

$$\sigma_{N,v}{}^2 = \left(\frac{\partial N_v}{\partial N_s}\sigma_{N,s}\right)^2 + \left(\frac{\partial N_v}{\partial M}\sigma_M\right)^2 + \left(\frac{\partial N_v}{\partial N_{s,0}}\sigma_{N,s,0}\right)^2 + \left(\frac{\partial N_v}{\partial M_0}\sigma_{M,0}\right)^2 \qquad (12)$$
$$+ \left(\frac{\partial N_v}{\partial N_{v,0,CTM}}\sigma_{N,v,0,CTM}\right)^2$$

$$\sigma_{N,v}{}^2 = \frac{1}{M^2}\left(\sigma_{N,s}{}^2 + \frac{(\Delta N_S + M_0 N_{v,0,CTM})^2}{M^2}\sigma_M{}^2 + \sigma_{N,s,0}{}^2 + N_{v,0,CTM}{}^2\sigma_{M,0}{}^2 \qquad (13) \right.$$
$$\left. + M_0{}^2\sigma_{N,v,0,CTM}{}^2\right)$$

where $\sigma_{N,s}$, $\sigma_M$, $\sigma_{N,s,0}$, $\sigma_{M,0}$ and $\sigma_{N,v,0,CTM}$ are respectively the errors on the slant column, the air mass factor,
and the slant column correction, the air mass factor, and the model vertical column in the reference sector
(indicated by suffix 0). For each of these categories, the following sections provide more details on the
implementation of the uncertainty estimate in the HCHO algorithm. A discussion of the sources of uncertainties
and, where possible, their estimated size are presented, as well as their spatial and temporal patterns.
Note that in the current implementation of the operational processor, $M_0 = M$, and the uncertainty formulation
therefore reduces to:



$$\sigma_{N,v}{}^2 = \frac{1}{M^2}\left(\sigma_{N,s}{}^2 + \frac{\Delta N_S{}^2}{M^2}\sigma_M{}^2 + \sigma_{N,s,0}{}^2\right) + \sigma_{N,v,0,CTM}{}^2 \tag{14}$$

Complementing this error propagation analysis, total column averaging kernels (*A*) based on the formulation
of Eskes and Boersma (2003) are estimated. Column averaging kernels provide essential information when
comparing measured columns with e.g. model simulations or correlative validation data sets, because they
allow removing the effect of the a-priori HCHO profile shape used in the retrieval (see APPENDIX C:
Averaging Kernel, Boersma et al., 2004; 2016).
Section 3.2 presents our current estimates of the precision (random uncertainty) and the trueness (systematic
uncertainty) that can be expected for the TROPOMI HCHO vertical columns. They are discussed along with
the product requirements (Section 2.1).
**3.1.1 Errors on the slant columns**
Error sources that contribute to the total uncertainty on the slant column originate both from instrument
characteristics and from uncertainties in the DOAS slant column fitting procedure itself.
The retrieval noise for individual observations is limited by the SNR of the spectrometer measurements. A
good estimate of the random variance of the reflectance (which results from the combined noise of radiance
and reference spectra) is given by the reduced $\chi^2$ of the fit, which is defined as the sum of squares (4) divided
by the number of degrees of freedom in the fit. The covariance matrix ($\Sigma$) of the linear least squares parameter
estimate is then given by:

$$\Sigma = \frac{\chi^2}{(k-n)}(A^T A)^{-1} \tag{15}$$

where *k* is the number of spectral pixels in the fitting interval, *n* is the number of parameters to fit and the
matrix $A(j \text{ x } k)$ is formed by the cross-sections. For each absorber *j*, the value $\sigma_{N,s,j}$ is usually called the slant
column error (SCE or $\sigma_{N,s,\text{rand}}$).

$$\sigma_{N,s,j}^2 = \frac{\chi^2}{(k-n)}(A^T A)_{j,j}^{-1} \tag{16}$$

Equation (16) does not take into account systematic errors, that are mainly dominated by slit function and
wavelength calibration uncertainties, absorption cross-section uncertainties, by interferences with other species
($O_3$, BrO or $O_4$), or by uncorrected stray light effects. The choice of the retrieval interval can have a significant
impact on the retrieved HCHO slant columns. The systematic contributions to the slant column errors are
empirically estimated from sensitivity tests (see Table 7) and can be viewed as part of the structural uncertainty
(Lorente et al., 2017). However, remaining systematic offsets and zonal biases are greatly reduced by the
reference sector correction. All effects summed in quadrature, the various contributions are estimated to
account for an additional systematic error of 20% of the background-corrected slant column:



$$\sigma_{N,s,syst} = 0.2\Delta N_s \qquad (17)$$

The total error on slant columns is then:

$$\sigma_{N,s}{}^2 = \sigma_{N,s,rand}{}^2 + \sigma_{N,s,syst}{}^2 \qquad (18)$$

**Table 7: Summary of the different error sources considered in the HCHO slant column uncertainty**
**budget.**

| Error source | Parameter uncertainty | Estimated uncertainty on HCHO SCD | Evaluation method - reference |
|---|---|---|---|
| **Measurement noise** | S/N=800-1000 | $1\times10^{16}$ molec.cm$^{-2}$ (random) | Value derived for individual observations by error propagation; De Smedt et al., 2015; |
| **HCHO cross-section error** | Based on alternative cross-section datasets, offset and polynomial orders. | 9% | Mean values derived from sensitivity tests using GOME-2 and OMI data. |
| **O$_3$ cross-section error** | | 5% | |
| **BrO cross-section error** | | 5% | De Smedt et al., 2008; 2015 Hewson et al., 2013 Pinardi et al., 2013 |
| **NO$_2$ cross-section error** | | 3% | |
| **Ring correction error** | | 5% | |
| **Choice of offset order** | | 7% | |
| **Choice of polynomial order** | | 7% | |
| **Instrumental slit function and wavelength calibration** | Based on alternative calibrations | 10% | Mean value derived from sensitivity tests using GOME-2 and OMI data. |
| **Choice of wavelength interval** | Based on alternative wavelength intervals | 10% | Mean value derived from sensitivity tests using GOME-2 and OMI data. Hewson et al., 2013 |
| **Temperature dependence of the HCHO XS** | 0.05%/°K | 2% | Mean value derived from sensitivity tests based on Meller and Moorgat (2000) |

**3.1.2 Errors on air mass factors**
The errors on the air mass factor depend on input parameter uncertainties and on the sensitivity of the air mass
factor to each of them. This contribution is broken down into the squared sum (Boersma et al., 2004, De Smedt
et al., 2008):

$$\sigma_M{}^2 = \left(\frac{\partial M}{\partial A_s} \cdot \sigma_{A,s}\right)^2 + \left(\frac{\partial M}{\partial f_c} \cdot \sigma_{f,c}\right)^2 + \left(\frac{\partial M}{\partial p_{cloud}} \cdot \sigma_{p,cloud}\right)^2 + \left(\frac{\partial M}{\partial s} \cdot \sigma_s\right)^2 + (0.2M)^2 \qquad (19)$$

The contribution of each parameter to the total air mass factor error depends on the observation conditions.
The air mass factor sensitivities ($M' = \frac{\partial M}{\partial parameter}$), i.e. the air mass factor derivatives with respect to the
different input parameters, can be derived for any particular condition of observation using the altitude-
dependent AMF LUT, and using the model profile shapes (see Figure 8). In practice, a LUT of AMF
sensitivities has been created using coarser grids than the AMF LUT, and one parameter describing the shape




of the profile: the profile height, i.e. the altitude (pressure) below which resides 75% of the integrated HCHO
profile. $\frac{\partial M}{\partial s}$ is approached by $\frac{\partial M}{\partial s_h}$ where $s_h$ is half of the profile height. Relatively small variations of this
parameter have a strong impact on the total air mass factors, because altitude-resolved air mass factors decrease
quickly in the lower troposphere, where the HCHO profiles peak (Figure 6).

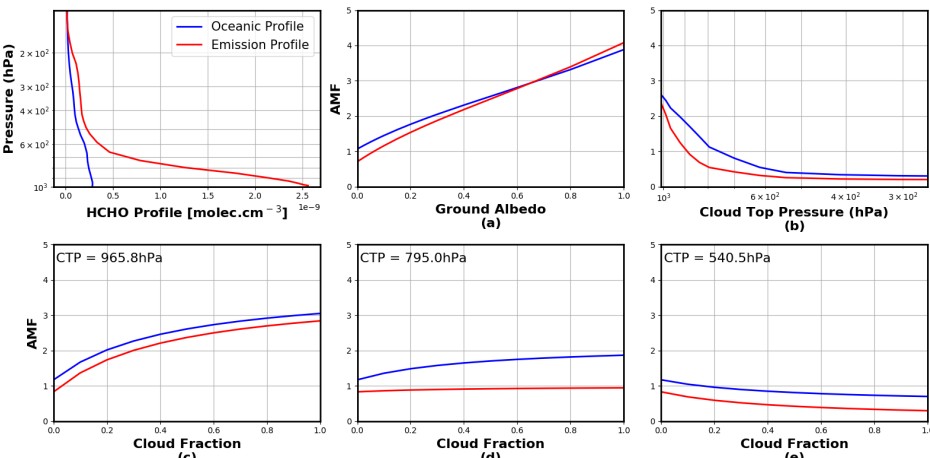


**Figure 8: First panel: TM5-MP HCHO profiles extracted in June over the equatorial Pacific ocean
(blue) and over Beijing (red). Those profiles have been used to calculated the tropospheric air mass
factors shown in the panels a to e, representing the AMF dependence on (a) the surface albedo, (b) the
cloud altitude, (c), (d), (e) the cloud fraction. In all cases, we consider a nadir view and a solar zenith
angle of 30°. In (a) the pixel is cloud free, in (b) the albedo is 0.02 and the effective cloud fraction is 0.5,
in (c), (d), (e) the ground albedo is 0.02 and the cloud pressure is respectively 966, 795 and 540 hPa.**

The errors $\sigma_{A,s}, \sigma_{f,c}, \sigma_{p,cloud}, \sigma_{s,h}$ are typical uncertainties on the surface albedo, cloud fraction, cloud top
pressure and profile shape, respectively. They are estimated from the literature or derived from comparisons
with independent data (see Table 8). Together with the sensitivity coefficients, these give the first four
contributions on the right of equation (19). The fifth term on the right of equation (19) represents the uncertainty
contribution due to possible errors in the AMF model itself (Lorente et al., 2017). We estimate this contribution
to 20% of the air mass factor (see also section 3.2.2).
Estimates of the air mass factor uncertainties and of their impact on the vertical column uncertainties are listed
in Table 8 and represented in Figure 9. They are based on the application of equation (19) to HCHO columns
retrieved from OMI measurements. In expression (19), the impact of possible correlations between
uncertainties on parameters is not considered, like for example the surface albedo and the cloud top pressure.
Note also that errors on the solar angles, the viewing angles and the surface pressure are supposed to be
negligible, which is not totally true in practice, since equation (10) does not yield the true surface pressure but
only a good approximation.





**Table 8: Summary of the different error sources considered in the air mass factor uncertainty budget.**

| Input parameter error | Symbol | Parameter Uncertainty | Source | Estimated uncertainty on HCHO VCD |
|---|---|---|---|---|
| Surface albedo | $\sigma_{A_s}$ | 0.02 | Kleipool et al., 2008 | 10-20% |
| Cloud fraction | $\sigma_{f,c}$ | 0.05 | Veefkind et al., 2016 | 05-15% |
| Cloud height | $\sigma_{p,cloud}$ | 50hPa | | 10-20% |
| Profile shape height | $\sigma_s$ | 100hPa | Upper limit of TM5-MP profile height standard deviation. | 20-60% |
| AMF wavelength dependency | Model / Structural uncertainty | 20% | Lorente et al., 2017 | 15-35% |
| LUT interp. errors | | | | |
| Model atmosphere | | | | |
| Cloud model/cloud correction/ | | | | |
| No explicit aerosol correction | | | | |


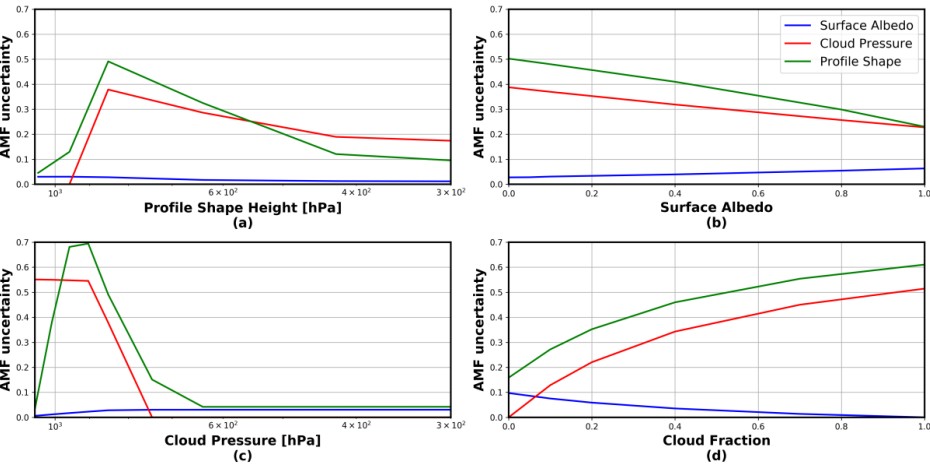


**Figure 9: AMF uncertainty related to profile shape, cloud pressure and surface albedo errors, as a**
**function of different observation conditions.**

**Surface albedo**
A reasonable uncertainty on the albedo is 0.02 (Kleipool et al., 2008). This translates to an error on the air mass
factor using the slope of the air mass factor as a function of the albedo and can be evaluated for each satellite
pixel (equation (19)). As an illustration, Figure 8 (a) shows the air mass factor dependence on the ground albedo
for two typical HCHO profile shapes (in blue: remote profile, in red: emission profile). At 340nm, the AMF





sensitivity (the slope), is almost constant with albedo, being only slightly higher for low albedo values. As
expected, the AMF sensitivity to albedo is higher for an emission profile peaking near the surface than for a
background profile more spread in altitude. More substantial errors can be introduced if the real albedo differs
considerably from what is expected, for example in the case of the sudden snowfall or ice cover. Snow/ice
cover map will therefore be used for flagging such cases.
**Clouds and aerosols**
An uncertainty on the cloud fraction of 0.05 is considered, while an uncertainty on the cloud top pressure of
50hPa is taken. Figure 8 (b) shows the air mass factor variation with cloud altitude. The AMF is very sensitive
to the cloud top pressure (the slope is steepest) when the cloud is located below or at the level of the
formaldehyde peak. For higher clouds, the sensitivity of the air mass factor to any change in cloud pressure is
very weak. As illustrated in Figure 8 (c), (d) and (e), for which a cloud top pressure of 966, 795 and 540 hPa
is respectively considered, the sensitivity to the cloud fraction is mostly significant when the cloud lies below
the HCHO layer.
The effect of aerosols on the air mass factors are not explicitly considered in the HCHO retrieval algorithm.
To a large extent, however, the effect of the non-absorbing part of the aerosol extinction is implicitly included
in the cloud correction (Boersma et al., 2011). Indeed, in the presence of aerosols, the cloud detection algorithm
is expected to overestimate the cloud fraction. Since non-absorbing aerosols and clouds have similar effects on
the radiation in the UV-visible range, the omission of aerosols is partly compensated by the overestimation of
the cloud fraction, and the resulting error on air mass factor is small, typically below 15% (Millet et al., 2006;
Boersma et al., 2011; Lin et al., 2014; Castellanos et al., 2015; Chimot et al. 2015). In some cases, however,
the effect of clouds and aerosols will be different. For example, when the cloud height is significantly above
the aerosol layer, clouds will have a shielding effect while the aerosol amplifies the signal through multiple
scattering. This will result in an underestimation of the AMF. Absorbing aerosols have also a different effect
on the air mass factors, since they tend to decrease the sensitivity to HCHO concentration. In this case, the
resulting error on the air mass factor can be as high as 30% (Palmer et al., 2001; Martin et al., 2002). This may,
for example, affect significantly the derivation of HCHO columns in regions dominated by biomass burning
as well as over heavily industrialized regions. Shielding and reflecting effect can thus occur, depending on the
observation, decreasing or increasing the sensitivity to trace gas absorption. It has been shown that uncertainties
related to aerosols is reduced by spatiotemporal averaging (Barkley et al., 2012; Lin et al., 2014; Castellanos
et al., 2015; Chimot et al. 2015). Furthermore, the applied cloud filtering effectively removes observations with
the largest aerosol optical depth. In the HCHO product, observations with an elevated absorbing aerosol index
will be flagged, to be used with caution.

**Profile shape**
This contribution to the total AMF error is the largest when considering monthly averaged observations. This
is supported by validation results using MAX-DOAS profiles measured around Beijing and Wuxi (see De
Smedt et al. 2015, Wang et al., 2016). Taking into account the averaging kernels allows removing from the





comparison the error related to the a priori profiles, when validating the results against other modelled or
measured profiles (see the APPENDIX C: Averaging Kernel).
**3.1.3 Errors on the reference sector correction**

$$\sigma_{N,v,0}{}^2 = \frac{1}{M^2}\left(\sigma_{N,s,0}{}^2 + N_{v,0,CTM}{}^2\sigma_{M,0}{}^2 + M_0{}^2\sigma_{N,v,0,CTM}{}^2\right) \tag{20}$$

This error includes contributions from the model background vertical column, from the error on the air mass
factor in the reference sector, and from the amplitude of the normalization applied to the HCHO columns. As
mentioned in 3.1.1, we consider that $\sigma_{N,s,0}$ is taken into account in Equation (17). The error on the air mass
factor in the reference sector $\sigma_{M,0}$ is calculated as in Equation (20) and saved during the background correction
step. Uncertainty on the model background has been estimated as the monthly averaged differences between
two different CTM simulations in the reference sector: IMAGES (Stavrakou et al., 2009a) and TM5-MP
(Huijnen et al., 2010). The differences range between 0.5 and 1.5x10$^{15}$ molec.cm$^{-2}$.
**Table 9: Estimated errors on the reference sector correction.**

| Error source | Uncertainty on HCHO VCD | Evaluation method – reference |
|---|---|---|
| Model background | 0.5 and 1.5x10$^{15}$ molec.cm$^{-2}$ | Difference between IMAGES and TM model |
| Amplitude of the column normalisation ($N_{s,0}$) | 0 to 4x10$^{15}$ molec.cm$^{-2}$ | Sensitivity tests using GOME-2 and OMI data. |

**3.2 HCHO error estimates and product requirements**
This section presents estimates of the precision (random error) and trueness (systematic error) that can be
expected for the TROPOMI HCHO vertical columns. These estimates are given in different NMVOC emission
regions. Precision and trueness of the HCHO product are discussed against the user requirements.
**3.2.1 Precision**
When considering individual pixels, the total uncertainty is dominated by the random error on the slant
columns. Our simulations and tests on real satellite measurements show that the precision by which the HCHO
can be measured is well defined by the instrument signal-to-noise level. For the nominal SNR level (1000), the
expected precision of single-pixel measurements is equivalent to the precision obtained with OMI HCHO
retrievals (De Smedt et al., 2015), but with a ground pixel size of about 3.5x7 km², i.e. one order of magnitude
smaller in surface. Absolute $\sigma_{N,s,rand}$ values typically range between 7 and 12x10$^{15}$ molec.cm$^{-2}$ for individual
pixels, showing an increase as a function of the surface altitude and of the solar zenith angle. Relative values
range between 100 and 300%, depending on the observation scene. In the case of HCHO retrievals, for
individual satellite ground pixels, the random uncertainty on the slant columns is the most important source of
uncertainty on the total vertical column. It can be reduced by averaging the observations, but of course at the
expense of a loss in time and/or spatial resolution.





The precision of the vertical columns provided in the L2 files corresponds to the precision of the slant column
divided by the air mass factor
$\sigma_{N,v,rand} = \frac{\sigma_{N,s,rand}}{M}$ (see Table 13).   It is dependent on the air mass factors, and therefore on the observation
conditions and on the cloud statistics. Figure 10 shows the vertical column precision that is expected for
TROPOMI, based on OMI observations in 2005. Results are shown in several regions, and at different spatial
and temporal scales (from individual pixels to monthly averaged column in 20x20km² grids). The product
requirements for HCHO measurements state a precision of $1.3 \times 10^{15}$ molec.cm$^{-2}$. This particular requirement
cannot be achieved with individual observations at full spatial resolution. However, as represented in Figure
10, the requirement can be approached using daily observations at the spatial resolution of 20x20km$^2$ (close to
the OMI resolution) or using monthly averaged columns at the TROPOMI resolution.  The precision can be
brought below $1 \times 10^{15}$ molec.cm$^{-2}$ if a spatial resolution of 20x20km$^2$ is considered for monthly averaged
columns.

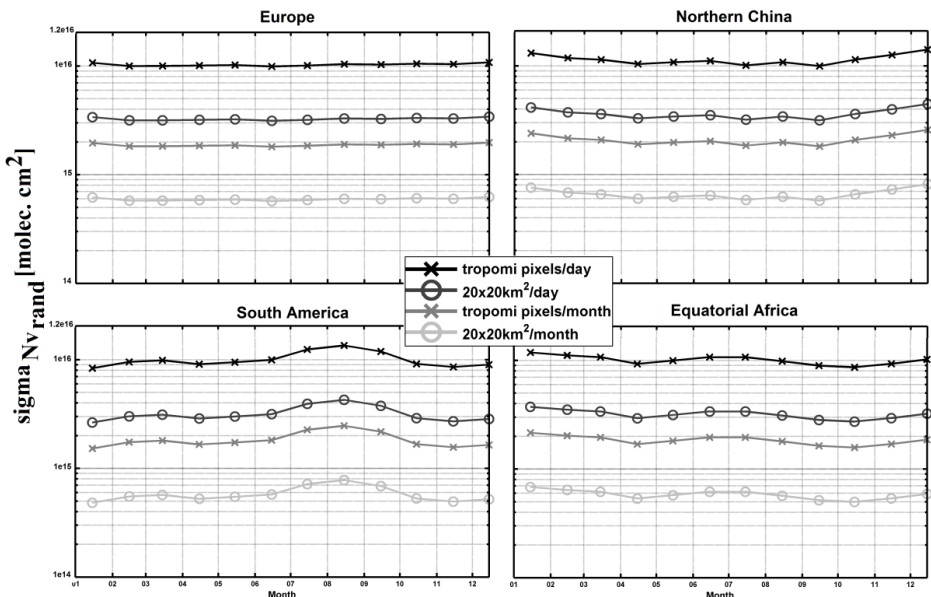


**Figure 10: Estimated precision on the TROPOMI HCHO columns, in several NMVOC emission**
**regions, and at different spatial and temporal scales (from individual pixels to monthly averages in**
**20x20 km² grids). These estimated are based on OMI observations in 2005, using observations with an**
**effective cloud fraction lower than 40%.**
**3.2.2 Trueness**
In this section, we present monthly averaged values of the systematic vertical columns uncertainties estimated
for OMI retrievals between 2005 and 2014. The contribution of the air mass factor uncertainties is the largest
contribution to the vertical column systematic uncertainties (see also Table 10). Figure 11 presents the VCD
uncertainties due to AMF errors, and the five considered contributions, over Equatorial Africa and Northern
China, as example of Tropical and mid-latitude sites. The largest contributions are from the a priori profile



uncertainty and from the structural uncertainty (taken as 20% of the AMF). In the case where the satellite
averaging kernels are used for comparisons with external HCHO columns, the a priori profile contribution can
be removed from the comparison uncertainty budget, leading to a total uncertainty in the range of 25% to 50%.
Table 10 wraps up the estimated relative contributions to the HCHO vertical column uncertainty, in the case
of monthly averaged columns for typical low and high columns.
Considering these estimates of the HCHO column trueness, the requirements for HCHO product (30%) are
achievable in regions of high emissions and for certain times of the year. In any case, observations need to be
averaged to reduce random uncertainties at a level comparable or smaller than systematic uncertainties.

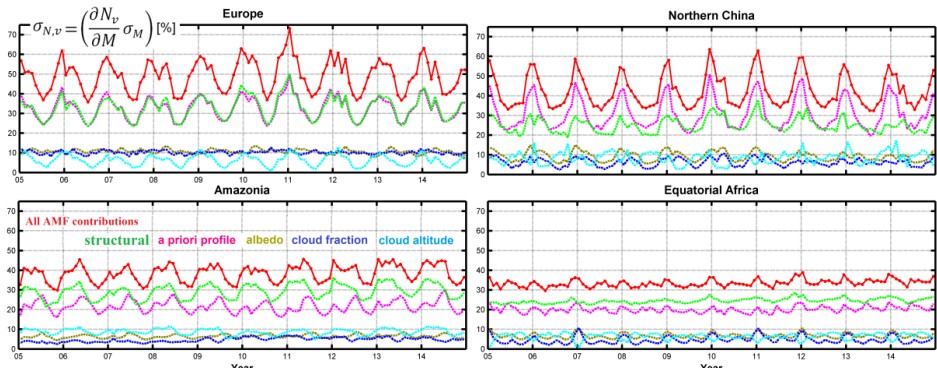


**Figure 11: Regional and monthly average of the relative systematic vertical column AMF-related**
**uncertainties in several NMVOC emission regions, for the period 2005-2014. The 5 contributions to the**
**systematic air mass factor uncertainty are shown: structural (green), a priori profile (pink), albedo**
**(olive), cloud fraction (blue) and cloud altitude (cyan).**
**Table 10: Estimated HCHO vertical column uncertainty budget for monthly averaged low and**
**elevated columns (higher than 1x10$^{16}$ molec.cm$^{-2}$). Contributions from the three retrieval steps are**
**provided, as well as input parameter contributions.**

| HCHO vertical error uncertainty | Remote regions / low columns | Elevated column regions / periods |
|---|---|---|
| Contribution from systematic slant columns uncertainties | 25% | 15% |
| Contribution from air mass factors uncertainties | 75% | 30% |
| • from a priori profile errors | • 60% | • 20% |
| • from model errors | • 35% | • 15% |
| • from albedo errors | • 20% | • 10% |
| • from cloud top pressure errors | • 20% | • 10% |
| • from cloud fraction errors | • 15% | • 05% |
| Contribution from background correction uncertainties | 40% | 10% |
| **Total** | **90%** | **35%** |
| **Total without smoothing error** | **50%** | **25%** |





## 4. Verification

In the framework of the TROPOMI L2 WG and QA4ECV projects, extensive comparisons of the prototype (this paper), the verification (IUP-UB), and alternative scientific algorithms (MPIC, KNMI, WUR) have been conducted. All follow a common DOAS approach. Prototype and verification algorithms have been applied to both synthetic and OMI spectra. Here, we present a selection of OMI results. For a complete description of the verification algorithm as well as results and discussion of the retrievals applied to synthetic spectra, please refer to the TROPOMI verification report (Richter et al., 2015).

### 4.1 Harmonized DOAS fit settings using OMI test data

For this exercise, a common set of DOAS fit parameters has been agreed upon. The goal of the intercomparison of harmonized fit settings was to ensure that the software implementation of the different algorithms behaves as expected in a large range of realistic measurement scenarios. Another objective was to gain knowledge on the level of agreement/disagreement of results from different groups when using the same settings, as well as on the main drivers for differences. Common and simple fit parameters based on the operational and verification algorithm were selected. They are summarized in Table 11.

**Table 11: Common DOAS fit settings for HCHO using OMI data.**

| Parameter | Values |
| --- | --- |
| **Fitting interval-1** | **328.5-359 nm** |
| **Calibration** | 1 interval (328-359 nm), using the SAO 2010 solar atlas (Chance and Kurucz, 2010). |
| **Molecular species** | HCHO, $NO_2$, Ozone, BrO, $O_2$-$O_2$ : same cross-sections as in Table 4 |
| **Ring effect** | Ring cross-section based on the technique outlined by Chance et al. (1997) |
| **Slit function** | One slit function per binned spectrum as a function of wavelength (60 OMI ISRF, Dirksen et al., 2006). |
| **Polynomial** | $5^{th}$ order |
| **Intensity offset correction** | Linear offset ($1/I_0$) |
| **Reference spectrum $I_0$** | Daily solar irradiance |

The intercomparison of results using common settings allowed to identify and fix several issues in the different codes leading to an overall consolidation of the algorithms. It has been found that minor changes in the fit settings may lead to large offsets ($\pm 10 \times 10^{15}$ molec.cm$^{-2}$) in the HCHO SCDs. However, an excellent level of agreement ($\pm 2 \times 10^{15}$ molec.cm$^{-2}$) between the different retrieval codes was obtained after several iterations of the common settings. The main sources of discrepancies were found to be related to (1) the solar $I_0$ correction applied on the $O_3$ cross-sections, (2) the intensity offset correction, (3) the details of the wavelength calibration of the radiance and irradiance spectra, and (4) the OMI slit functions and their implementation in the convolution tools (Boersma et al., 2015).



An overview of the final SCD comparison is shown on Figure 12 for six test days at the beginning and the end
of the OMI time series, and for a particular OMI orbit on the left panel of Figure 13. The correlation coefficient,
slope and offset of linear regression fits performed on each comparison orbit are displayed. The correlation of
slant columns from BIRA and IUP-UB is extremely high in most cases. It is > 0.998 for all orbits. The slope
of the regression line between BIRA and IUP-UB results is close to 1.0. There is a constant offset of less than
$1 \times 10^{15}$ molec.cm$^{-2}$. The comparison between MPIC results and the two other algorithms gives somehow lower
correlations, but still larger than 0.98 from the beginning to the end of the OMI lifetime. Final deviations on
OMI HCHO SCD when using common settings were found to be of maximum +-2% (slope) and $2.5 \times 10^{15}$
molec.cm$^{-2}$. When relating the remaining differences in retrieved SCDs using common settings to the slant
column errors from the DOAS fit ($\sigma_{N,s,rand}$), it can be concluded that the differences between the results are
significantly smaller than the uncertainties (from 10 to 20% of $\sigma_{N,s,rand}$). Moreover, remaining offsets in SCDs
are further reduced by the background correction procedure.

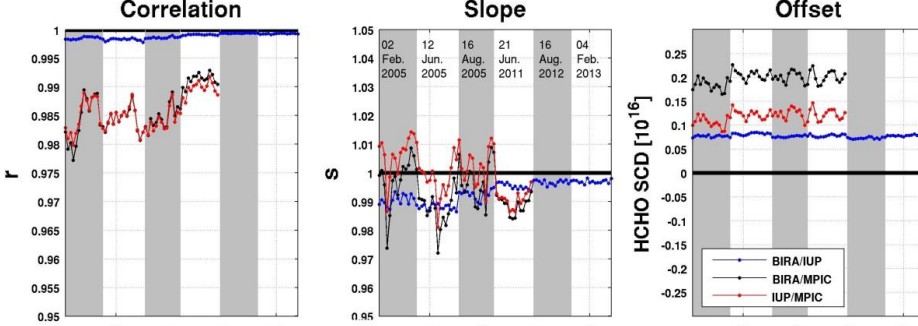


**Figure 12: Correlation (left), slope (middle) and offset (right) from a linear regression performed for**
**the common fit settings (see Table 11) for each orbit of OMI test days. A correlation plot for an**
**example orbit is provided in the left panel of Figure 13**Error! Reference source not found.**.**
**4.2 Verification of the operational implementation**
A similar intercomparison exercise was performed with the operational algorithm UPAS, developed at DLR,
but using the exact settings of the prototype algorithm as detailed in Table 2. An example of resulting
correlation fit is shown in the right panel of Figure 13 for the same OMI orbit as for the comparison with the
IUP-UB results. The level of agreement between the prototype and operational results is found to be almost
perfect (correlation coefficient of 1, slope of 1.003 and offset of less than $0.2 \times 10^{15}$ molec.cm$^{-2}$), and very
satisfactory considering the sensitivity on small implementation changes.





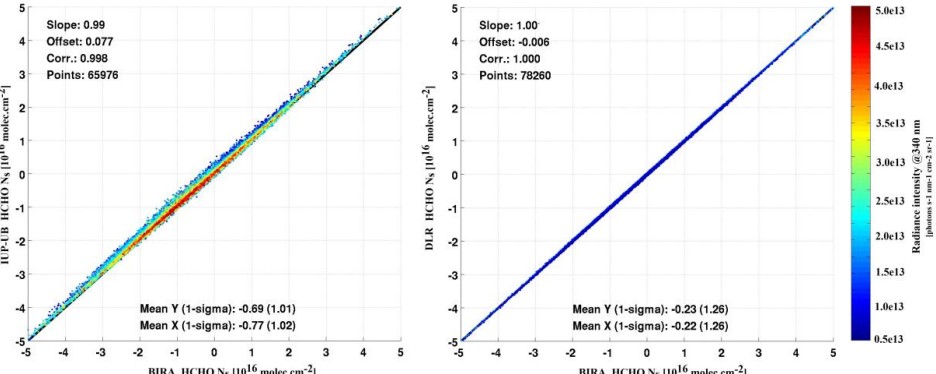

**Figure 13: Correlation plots of HCHO slant columns retrieved with the BIRA prototype algorithm and (left) the IUP-UB verification algorithm, (right) the operational processor, for OMI orbit number 2339 on 02/02/2005, including all pixels with SZA<80°.**

## 5. Validation

Independent validation activities are proposed and planned by the S5P Validation Team (Fehr, 2016) and within the ESA S5P Mission Performance Center (MPC). The backbone of the formaldehyde validation is the MAX-DOAS and FTIR networks operated as part of the Network for the Detection of Atmospheric Composition Change (NDACC, www.ndsc.ncep.noaa.gov/) complemented by PANDONIA (pandonia.net/) and national activities. In addition, model datasets will be used for validation as well as independent satellite retrievals. Finally, airborne campaigns are planned to support the formaldehyde and other trace gases validation.

### 5.1 Requirements for validation

To validate the TROPOMI formaldehyde data products, comparisons with independent sources of HCHO measurements are required. This includes comparisons with ground-based measurements, aircraft observations and satellite data sets from independent sensors and algorithms. Moreover, not only information on the total (tropospheric) HCHO column is needed but also information on its vertical distribution, especially in the lowest three kilometres where the bulk of formaldehyde generally resides. In this altitude range, the a-priori vertical profile shapes have the largest systematic impact on the satellite column errors. HCHO and aerosol profile measurements are therefore needed.

The diversity of the NMVOC species, lifetimes and sources (biogenic, biomass burning or anthropogenic) calls for validation data in a large range of locations worldwide (tropical, temperate and boreal forests, urban and sub-urban areas). Continuous measurements are needed to obtain good statistics (as well for ground-based measurements as for satellite columns) and to capture the seasonal variations. Validation and assessment of consistency with historical satellite datasets require additional information on the HCHO diurnal variation, which depends on the precursor emissions and on the local chemical regime.



The main emphasis is on quality assessment of retrieved HCHO column amounts on a global scale and over
long time periods. The validation exercise will establish whether HCHO data quality meets the requirements
of geophysical research applications like long term trend monitoring on the global scale, NMVOC source
inversion, and research on the budget of tropospheric ozone. In addition, the validation will investigate the
consistency between TROPOMI HCHO data and HCHO data records from other satellites.
**5.2 Reference measurement techniques**
Table 12 summarizes the type of data and measurements that can be used for the validation of the TROPOMI
HCHO columns. The advantages and limitations of each technique are discussed. It should be noted that, unlike
tropospheric $O_3$ or $NO_2$, the stratospheric contribution to the total HCHO column can be largely neglected
which simplifies the interpretation of both satellite and ground-based measurements.
**Table 12: Data/Measurement types used for the validation of satellite HCHO columns. The**
**information content of each type of measurement is qualitatively represented by the number of crosses.**

| Type of measurement | Sensitivity in the boundary layer | Vertical profile information | Diurnal variation | Seasonal Variation | Total column | Earth coverage |
|---|---|---|---|---|---|---|
| MAX-DOAS | xxx | xx (3) | xxx | xxx | xx | xx |
| FTIR | x | - | xxx | xx | xxx | x |
| Direct Sun | xxx | - | xxx | xxx | xxx | x |
| In situ (1) | xx | - | xxx | xxx | - | xx |
| Aircraft (2) | xx | xxx | x | - | xx (4) | x |
| Satellite instruments | x | - | x | xxx | xx | xxx (5) |


(1) Surface measurements that could be combined with regional modelling.
(2) Including ultra-light and unmanned airborne vehicles.
(3) Up to 2-3 km.
(4) Profiles generally need to be extrapolated.
(5) Different daily coverage and spatial resolutions.
The Multi-axis DOAS (MAX-DOAS) measurement technique has been developed to retrieve stratospheric and
tropospheric trace gas total columns and profiles. The most recent generation of MAX-DOAS instruments
allows for measurement of aerosols and a number of tropospheric pollutants, such as $NO_2$, HCHO, $SO_2$, $O_4$
and CHOCHO (e.g. Irie et al., 2011). With the development of operational networks such as Pandonia
(http://pandonia.net/), it is anticipated that many more MAX-DOAS instruments will become available in the
near future to extend validation activities in other areas where HCHO emissions are significant. The locations
where HCHO measurements are required are reviewed in the next section. Previous comparisons between
GOME-2 and OMI HCHO monthly averaged columns with MAX-DOAS measurements recorded by BIRA-
IASB in the Beijing city centre and in the sub-urban site of Xianghe showed that the systematic differences
between the satellite and ground-based HCHO columns (about 20 to 40%) are almost completely explained



when taking into account the vertical averaging kernels of the satellite observations (De Smedt et al., 2015,
Wang et al., 2017), showing the importance of validating the a priori profiles as well.
HCHO columns can also be retrieved from the ground using FTIR spectrometers. In contrast to MAXDOAS
systems which essentially probe the first two kilometres of the atmosphere, FTIR instruments display a strong
sensitivity higher up in the free troposphere and are thus complementary to MAXDOAS (Vigouroux et al.,
2009). The deployment of FTIR instruments of relevance for HCHO is mostly taking place within the NDACC
network. Within the project NIDFORVal (S5P Nitrogen Dioxide and Formaldehyde Validation using NDACC
and complementary FTIR and UVVis networks), the number of FTIR stations providing HCHO time-series
has been raised from only 4 (Vigouroux et. al, 2009; Jones et al., 2009; Viatte et al., 2014; Franco et al., 2015)
to 21. These stations are covering a wide range of HCHO concentrations, from clean Arctic or oceanic sites to
sub-urban and urban polluted sites, as well as sites with large biogenic emissions such as Porto Velho (Brazil)
or Wollongong (Australia).
Although ground-based remote-sensing DOAS and FTIR instruments are naturally best suited for the validation
of column measurements from space, in-situ instruments can also bring useful information. This type of
instrument can only validate surface HCHO concentrations, and therefore additional information on the vertical
profile (e.g. from regional modelling) is required to make the link with the satellite retrieved column. However,
in-situ instruments (where available) have the advantage to be continuously operated for pollution monitoring
in populated areas, allowing for extended and long term comparisons with satellite data (see e.g. Dufour et al.,
2009). Although more expensive and with a limited time and space coverage, aircraft campaigns provide
unique information on the HCHO vertical distributions (Zhu et al., 2017).
**5.3 Deployment of validation sites**
Sites operating correlative measurement should preferably be deployed at locations where significant NMVOC
sources exist. This includes:
• Tropical forests (Amazonian forest, Africa, Indonesia): The largest HCHO columns worldwide are
observed over these remote areas that are difficult to access. Biogenic and biomass burning emissions are
mixed. A complete year is needed to discriminate the various effects on the HCHO retrieval. Clouds tend
to have more systematic effects in tropical regions. Aircraft measurements are needed over biomass
burning areas.
• Temperate forests (South-Eastern US, China, Eastern Europe): In summer time, HCHO columns are
dominated by biogenic emissions. Those locations are useful to validate particular a-priori assumptions
such as model isoprene chemistry and OH oxidation scheme. Measurements are mostly needed from April
to September.
• Urban and sub-urban areas (Asian cities, California, European cities): Anthropogenic NMVOCs are more
diverse, and have a weaker contribution to the total HCHO column than biogenic NMVOCs. This type of
signal is therefore more difficult to validate. Continuous observations at mid-latitudes over a full year are
needed, to improve statistics.



For adequate validation, the long-term monitoring should be complemented by dedicated campaigns. Ideally
such campaigns should be organised in appropriate locations such as e.g. South-Eastern US, Alabama where
biogenic NMVOCs and biogenic aerosols are emitted in large quantities during summer time, and should
include both aircraft and ground-based components.
**5.4 Satellite-satellite intercomparisons**
Satellite-satellite intercomparisons of HCHO columns are generally more straightforward than validation using
ground-based correlative measurements. Such comparisons are evaluated in a meaningful statistical sense
focusing on global patterns and regional averages, seasonality, scatter of values and consistency between
results and reported uncertainties. When intercomparing satellite measurements, special care has to be drawn
to:
• differences in spatial resolutions, resulting in possible offsets between satellite observations (van
der A et al., 2008; De Smedt et al., 2010; Hilboll et al., 2013),
• differences in overpass times, that holds valuable geophysical information about diurnal cycles
in emissions and chemistry (De Smedt et al., 2015; Stavrakou et al., 2015)
• differences in a priori assumptions.
• differences in the cloud algorithms and cloud correction schemes.
Assessing the consistency between successive satellite sensors is essential to allow for scientific studies making
use of the combination of several sensors. For example trends in NVMOC emissions have been successfully
derived from GOME(-2), SCIAMACHY, and OMI measurements (Figure 14). It is anticipated that TROPOMI,
the next GOME-2 instruments and the future Sentinel-4 and -5, will allow to extend these time series.

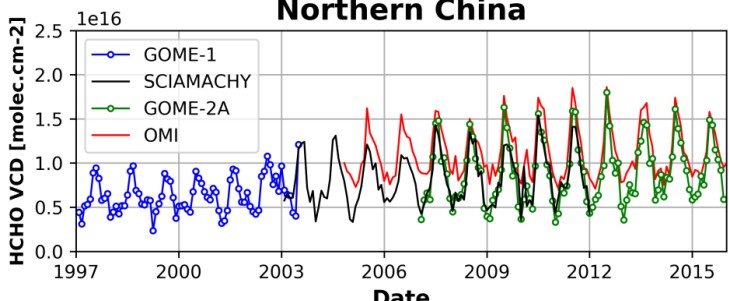


**Figure 14: HCHO columns over Northern China as observed with GOME (in blue), SCIAMACHY (in**
**black), GOME-2 (in green), and OMI (in red) (De Smedt et al., 2008; 2010; 2015).**



## 6. Conclusions

The retrieval algorithm for the TROPOMI formaldehyde product generation is based on the heritage from algorithms successfully developed for the GOME, SCIAMACHY, GOME-2 and OMI sensors. A double-interval fitting approach is implemented, following an algorithm baseline demonstrated on the GOME-2 and OMI sensors. The HCHO retrieval algorithm also includes a post-processing across-track reference sector correction to minimize OMI-type striping effects, if any. Additional features for future processor updates include the use of daily earthshine radiance as reference selected in the remote Pacific spectral, outlier screening during the fitting procedure (spike removal algorithm), and a more accurate background correction scheme.

A detailed uncertainty budget is provided for every satellite observation. The precision of the HCHO tropospheric column is expected to come close to the COPERNICUS product requirements in regions of high emissions and, at mid-latitude, for summer (high sun) conditions. The trueness of the vertical columns is also expected to be improved, owing to the use of daily forecasts for the estimation of HCHO vertical profile shapes, that will be provided by a new version of the TM5-MP model, running at the spatial resolution of 1x1 degree in latitude and longitude.

The validation of satellite retrievals in the lower troposphere is known to be challenging. Ground-based measurements, where available, often sample the atmosphere at different spatial and temporal scales than the satellite measurements, which leads to ambiguous comparisons. Additional correlative measurements are needed over a variety of regions, in particular in the Tropics and at the sub-urban level in mid-latitudes. These aspects are covered by a number of projects developed in the framework of the TROPOMI validation plan (Fehr, 2016).

**Acknowledgements**

The TROPOMI HCHO algorithmic developments have been supported by the ESA Sentinel-5 Precursor Level-2 Development project, as well as by the Belgian PRODEX (TRACE-S5P project). Multi-sensor HCHO developments have been funded by the EU FP7 QA4ECV project (grant no. 607405), in close cooperation with KNMI, University of Bremen, MPIC-Mainz and WUR.





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



**1200**    **APPENDIX A: Acronyms and abbreviations**

| | |
|---|---|
| A | Averaging Kernel |
| AMF | Air mass factor |
| AOD | Aerosol optical depth |
| AAI | Aerosol absorbing index |
| ATBD | Algorithm Theoretical Basis Document |
| BIRA-IASB | Royal Belgian Institute for Space Aeronomy |
| BrO | Bromine Monoxide |
| BRDF | Bidirectional reflectance distribution function |
| $CH_4$ | Methane |
| CO | Carbon Monoxide |
| CAPACITY | Composition of the Atmosphere: Progress to Applications in the user CommunITY |
| CCD | Charged Coupled Device |
| CF | Climate and Forecast metadata conventions |
| CRB | Clouds as Reflecting Boundaries |
| CTM | Chemical Transport Model |
| DOAS | Differential optical absorption spectroscopy |
| DU | Dobson Unit (1 DU = $2.6867 \times 10^{16}$ molecules $cm^{-2}$) |
| ECMWF | European Centre for Medium Range Weather Forecast |
| ESA | European Space Agency |
| FWHM | Full Width Half Maximum |
| GMES | Global Monitoring for Environment and Security |
| GOME | Global Ozone Monitoring Experiment |
| HCHO | Formaldehyde (or $H_2CO$) |
| IPA | Independent Pixel Approximation |
| IR | Infrared |
| ISRF | Instrument Spectral Response Function |
| L2 | Level-2 |
| L2WG | Level-2 Working Group |
| LER | Lambertian Equivalent Reflector |
| VLIDORT | Vector LInearized Discrete Ordinate Radiative Transfer |
| LOS | Line-of-sight angle |
| LS | Lower stratosphere |
| LUT | Look-up table |
| MAX-DOAS | Multi-axis DOAS |
| MPC | Mission Performance Center |
| NDACC | Network for the Detection of Atmospheric Composition Change |
| NMVOC | Non-Methane Volatile Organic Compound |
| $NO_2$ | Nitrogen Dioxide |



| NRT | Near-real time |
| OCRA | Optical Cloud Recognition Algorithm |
| OD | Optical Depth |
| $O_3$ | Ozone |
| OMI | Ozone Monitoring Instrument |
| OMPS | Ozone Mapping Profiler Suite |
| (P)BL | Planetary Boundary Layer |
| PCA | Principal Component Analysis |
| QA4ECV | Quality Assurance For Essential Climate Variables |
| RAA | Relative Azimuth Angle |
| ROCINN | Retrieval Of Cloud Information using Neural Networks |
| RRS | Rotational Raman Scattering |
| RTM | Radiative transfer model |
| S5P | Sentinel-5 Precursor |
| S5 | Sentinel 5 |
| SAA | Solar Azimuth Angle |
| SCIAMACHY | SCanning Imaging Absorption spectroMeter for Atmospheric ChartograpHY |
| SC(D) | Slant column density |
| SCDE | Slant column density error |
| SNR | Signal-to-noise ratio |
| $SO_2$ | Sulfur dioxide |
| SOW | Statement Of Work |
| SWIR | Short-wave infrared |
| SZA | Solar zenith angle |
| TM 4/5 | Data assimilation / chemistry transport model (version 4 or 5) |
| TROPOMI | Tropospheric Monitoring Instrument |
| UPAS | Universal Processor for UV/VIS Atmospheric Spectrometers |
| UV | Ultraviolet |
| UVN | Ultraviolet/Visible/Near-infrared |
| VAA | Viewing Azimuth Angle |
| VZA | Viewing Zenith Angle |
| VC(D) | Vertical column density |



**APPENDIX B: High level L2 HCHO data product description**
In addition to the main product results, such as HCHO slant column, tropospheric vertical column and air mass
factor, the level 2 data files contain a number of additional ancillary parameters and diagnostic information. A
complete description of the level 2 data format is given in the Product User Manual (Pedergnana et al., 2017).
A selection of important parameters is given in Table 13. A complete description of the level 2 data format is
given in the Product User Manual (Pedergnana et al., 2017).
**Table 13: Selective list of output fields in the TROPOMI HCHO product. Scanline and ground_pixel**
**are respectively the number of pixels in an orbit along track and across track. Layer is the number of**
**vertical levels in the averaging kernels and the a-priori profiles.**

| Symbol | Unit* | Variable name | Number of entries |
|--------|-------|---------------|-------------------|
| $N_v$ | mol.m$^{-2}$ | formaldehyde_tropospheric_vertical_column | scanline x ground_pixel |
| $N_s$ | mol.m$^{-2}$ | fitted_slant_columns | scanline x ground_pixel x number_of_slant_columns |
| $N_s - N_{s,0}$ | mol.m$^{-2}$ | formaldehyde_slant_column_corrected | scanline x ground_pixel |
| $N_{v,0}$ | mol.m$^{-2}$ | formaldehyde_tropospheric_vertical_column_correction | scanline x ground_pixel |
| $M$ | n.u. | formaldehyde_tropospheric_air_mass_factor | scanline x ground_pixel |
| $M_{clear}$ | n.u. | formaldehyde_clear_air_mass_factor | scanline x ground_pixel |
| $f_c$ | n.u. | cloud_fraction_crb | scanline x ground_pixel |
| $w_c$ | n.u. | cloud_fraction_intensity_weighted | scanline x ground_pixel |
| $p_{cloud}$ | Pa | cloud_pressure_crb | scanline x ground_pixel |
| $A_{cloud}$ | n.u. | cloud_albedo_crb | scanline x ground_pixel |
| $A_s$ | n.u. | surface_albedo | scanline x ground_pixel |
| $z_s$ | m | surface_altitude | scanline x ground_pixel |
| $\sigma_{N,v,rand}$ | mol.m$^{-2}$ | formaldehyde_tropospheric_vertical_column_precision | scanline x ground_pixel |
| $\sigma_{N,v,syst}$ | mol.m$^{-2}$ | formaldehyde_tropospheric_vertical_column_trueness | scanline x ground_pixel |
| $\sigma_{N,s,rand}$ | mol.m$^{-2}$ | fitted_slant_columns_precision | scanline x ground_pixel x number_of_slant_columns |
| $\sigma_{M,rand}$ | n.u. | formaldehyde_tropospheric_air_mass_factor_precision | scanline x ground_pixel |
| $\sigma_{N,s,0}$ | mol.m$^{-2}$ | formaldehyde_slant_column_corrected_trueness | scanline x ground_pixel |
| $A$ | n.u. | averaging_kernel | layer x scanline x ground_pixel |
| $n_a$ | vmr | formaldehyde_profile_apriori | layer x scanline x ground_pixel |
| $p_s$ | Pa | surface_pressure | scanline x ground_pixel |
| $a_l$ | Pa | tm5_constant_a | layer |



| Symbol | Unit* | Variable name | Number of entries |
|---|---|---|---|
| $b_l$ | n.u. | tm5_constant_b | layer |
| $N_{s,1}$ | mol.m$^{-2}$ | fitted_slant_columns_win1 | scanline x ground_pixel x number_of_slant_columns_win1 |
| $\sigma_{N,s,1,rand}$ | mol.m$^{-2}$ | fitted_slant_columns_precision_win1 | scanline x ground_pixel x number_of_slant_columns_win1 |

* multiplication factor to convert mol.m$^{-2}$ to molec.cm$^{-2}$: 6.022x10$^{19}$





**APPENDIX C: Averaging Kernel**
Retrieved satellite quantities always represent a weighted average over all parts of the atmosphere that
contribute to the signal observed by the satellite instrument. The DOAS total column retrieval is implicitly
dependant on the a priori trace gas profile $n_a$. Radiative transfer calculations account for the sensitivity of the
measurement to the HCHO concentrations at all altitudes and these sensitivities are weighted with the assumed
a priori profile shape to produce the vertical column. The averaging kernel ($A$) is proportional to the
measurement sensitivity profile, and provides the relation between the retrieved column $N_v$ and the true tracer
profile x (Rodgers, 2000; Rodgers and Connor, 2002):

$$N_v - N_{v,a} = A.(x^{pc} - n_a^{pc}) \tag{21}$$

where the profiles are expressed in partial columns ($pc$). For total column observations of optically thin
absorbers DOAS averaging kernels are calculated as follows (Eskes and Boersma, 2003): $A(p) = \frac{m(p)}{M}$, where
$m(p)$ is the altitude-resolved air mass factor and $M$ is the tropospheric air mass factor. The air mass factor and
therefore the retrieved vertical column, depends on the a priori profile shape, in contrast to the altitude-resolved
air mass factor which describes the sensitivity of the slant column to changes in trace gas concentrations at a
given altitude and does not depends on the a priori profile in an optically thin atmosphere. From the definition
of $A$, we have $N_{v,a} = A.n_a^{pc}$ and Equation (21) simplifies to:

$$N_v = A.x^{pc} \tag{22}$$

The averaging kernel varies with the observation conditions. In the HCHO retrieval product, $A$ is provided
together with the error budget for each individual pixel. The provided HCHO vertical columns can be used in
two ways, each with its own associated error (Boersma et al., 2004):
1.    For independent study and/or comparison with other independent measurements of total column amounts.

In this case, the total error related to the column consists of slant column measurement errors, reference

sector correction errors, and air mass factor errors. The latter consists of errors related to uncertainties in

the assumed profile $n_a$ and errors related to the $m$ parameters.

2.    For comparisons with chemistry transport models or validation with independent profile measurements,

if the averaging kernel information is used, the a priori profile shape error no longer contributes to the

total error. Indeed, the relative difference between the retrieved column $N_v$ and an independent profile $x_i$

is:

$$\delta = \frac{N_v - A.x_i^{pc}}{N_v} \tag{23}$$




The total AMF $M$ cancels since it appears as the denominator of both $N_v$ and $A$. Because only the total
AMF depends on the a priori tracer profile $n_a$, the comparison using the averaging kernel is not influenced
by the chosen a priori profile shape. The a priori profile error does not influence the comparison, but of
course, it still does influence the error on the retrieved vertical column.