# Peer review of "Algorithm Theoretical Baseline for formaldehyde retrievals"

_Atmospheric Measurement Techniques, 2017_

## Referee Comment (RC1) · Anonymous Referee #1 · 29 Dec 2017

De Smedt and co-authors describe in great detail the theoretical basis, methods and assumptions used in the operational processor of the Sentinel-5 Precursor TROPOspheric Monitoring Instrument (S5P TROPOMI). The algorithm described with great detail here benefits from strong heritage having the co-authors lead the development of formaldehyde retrievals using measurements from previous UV space sensors. Algorithm improvements developed during the Quality Assurance for Essential Climate Variables (QA4ECV) project, funded by the European Union, are also presented since they are the basis of future updates to the operational processor. The error budget for TROPOMI formaldehyde observations is derived and discussed in the context of the Copernicus requirements. Finally validation methods and goals are discussed.

This paper should be the reference document for TROPOMI formaldehyde. Anyone using the operational product should read it to understand the meaning of the retrieved quantities, and their suitability to carry on scientific studies. The content is presented in a clear and sound way, it follows the logical steps of the algorithm and is well organized. The paper is ready to be published with minor changes that will only add to its great value. A set of recommendations to minimally expand the content of the paper is followed by some technical corrections.

**Section 1. Introduction.**

Some extra references should be added to support the statements describing formaldehyde chemistry in the atmosphere.

**Section 2.2 Algorithm description**

Why are the HCHO Meller and Moorgat (2000) cross sections used instead of the more recent and intensity corrected Chance and Orphal (2011)?

**Section 2.2.1 Formaldehyde slant column retrieval**

Page 11, line 233. It will be valuable to show evidence of the reduction in the correlation between formaldehyde and bromine moxide by using a two-step DOAS retrieval adding a new figure? Given the extension of the paper it is maybe not necessary, but it will be interesting to have it here.

Page 11, line 264. The text says "(3) possible row-dependent biases (stripes) are directly corrected owing to the use of one reference per detector row." Are irradiances not recorded for each detector row? If they are, as it is done with OMI, the reason for the removal of the stripes when using radiance reference should be other than just having an irradiance reference for each row.

Page 12, line 284. Equation 5 only shows the shift ($\Delta_i$) but line 284 mentions that this approach allows compensating for stretch and shift errors. How correlated are those? Is it not possible to treat them separately?

**Section 2.2.2 Tropospheric air mass factor**

An evaluation of the effect of using only one atmospheric model (US Standard) for the calculation of altitude dependent air mass factors should be included. Ozone distribution can vary significantly between tropics, polar region and season.

Another subsection could be added to discuss the role of the surface reflectance properties to complete the description since all other AMF parameters have their own (LUT of altitude dependent AMFs, cloudy scenes, aerosols, and a priori vertical profile shapes).

**Section 2.2.3. Across-track and zonal reference sector correction**

Page 20, line 470. "The natural background level of HCHO is well estimated from chemistry model simulations of $CH_4$". Actually there is evidence that models underpredict HCHO in the Pacific Ocean (http://onlinelibrary.wiley.com/doi/10.1002/2016JD026121/abstract). Add a sentence discussing this situation.

As said above, despite the paper being fairly long, it could benefit of a plot showing the process of the background correction for one orbit illustrating the changes for each step.

**Section 3.1.1. Errors on the slant columns**

Please add $O_4$ uncertainty to table 7.

**Section 3.1.2. Errors on air mass factors**

Page 26, Figure 9: Would it be possible to specify the geometry, surface pressure, and the rest of parameters kept fix in the calculation for each panel.

Page 26, **Surface albedo**: Kleipool et al., surface climatology has a coarse resolution (0.5°x0.5°) compared with TROPOMI pixels. It would be interesting to incorporate in the error analysis the uncertainties associated with subpixel inhomogeneity in the Kleipool database or at least discuss them in the text.

Page 27, **Clouds and aerosols**: In section 2.2.2 "Tropospheric air mass factor" page 17, line 403 it is said that pixels with cloud fractions below 10% are considered clear-sky pixels "to avoid unnecessary error propagation through the retrievals" given the unstable cloud retrieval for such conditions. Under that assumption, how are the AMF errors due to cloud parameters calculated?

Page 27, **Profile shape**: As for the surface climatology, would it be possible to estimate the uncertainty derived from subpixel model inhomogeneity given that model information is available in 1° grid and TROPOMI pixels can be as small as 7x7 km²?

**Section 4. Verification**

Building on the work by Lorente et al., 2017 was the AMF calculation tested using harmonized parameters.

Page 34, Table 12: The number of xx in the Earth coverage column for MAX-DOAS and Direct Sun, should it not be similar given that most MAX-DOAS instruments can also carry on Direct Sun measurements?

Page 35, after paragraph devoted to MAX-DOAS could add a little paragraph describing Direct Sun capabilities.

It will be nice to add a map of current ground-based measurements sites lined up for validation.

**Section 5.4 Satellite-satellite intercomparisons**

Page 36, line 861: For completeness about current and future instruments it will be good to add mentions to OMPS, GEMS, and TEMPO.

**Technical Corrections:**

Page 1, line 33: "Its lifetime being of the order of a few hours, …" is grammatically incorrect. What about, "With its lifetime of the order of a few hours, HCHO concentrations in the boundary layer…"

Page 2, line 43: Would you consider to add Kaiser et al., 2017 to the list of inversion studies (https://doi.org/10.5194/acp-2017-1137)?

Page 2, line 52: To complete the list of HCHO retrievals from LEO it should be added the ones using OMPS measurements (http://onlinelibrary.wiley.com/doi/10.1002/2015GL063204/abstract, https://www.atmos-meas-tech.net/9/2797/2016/).

Page 3, line 77: Is there any reference or link available to the S5P HCHO Level 2 Algorithm Theoretical Basis Document v1.0

Page 4, Table 1: What is the meaning of revisit time 24x3 hour. Since h is used in the top line it will be good to make both units consistent (h or hour).

Page 5, Figure 2: Geolocation and Time information also need to feed the HCHO climatology or TM5 daily forecast.

Page 6, line 156: "Figure 3 also" would read better if just said "Figure 3 presents"

Page 8, Figure 3 caption: Mention that these vertical columns are derived using OMI data. It is said in the text in section 2.2.1 (line 246) but the first time figure 3 is referenced in the text, section 2.2 (line 156) nothing is mention and there may be misunderstandings.

Page 14, line 337: "average" should be "addition" or "sum".

Page 14, line 341: The symbols for solar zenith angle, viewing zenith angle, and relative azimuth angle are not defined in the text. Later on they are defined in table 4 but that only happens in page 15.

Page 16, line 385: "in which a inhomogeneous" should read "in which an inhomogeneous"

Page 19, line 449: Clarify that they are OMI air mass factor for example saying "Yearly averaged OMI air mass factors…"

Page 28, line 642: "Equation (20)" should be "Equation (19)"?

Page 29, Table 9. "0.5 and 1.5x10$^{15}$ molec.cm$^{-2}$" should be "0.5 to 1.5x10$^{15}$ molec.cm$^{-2}$"?

Page 32, Figure 12 caption: Check "Error! Reference source not found"

Page 33, line 774: suggest to change the text between brackets to "(both for ground-based measurements and for satellite columns)"

Page 35, line 827: "measurement" should read "measurements"?

Page 37, line 871: Move comma before spectral

Page 50, line 1206: Remove sentence "A complete description of the level 2 data…" since it's a repetition of the sentence in line 1203.

---

## Short Comment (SC1) · 12 Jan 2018

**Short comment: *Algorithm Theoretical Baseline for formaldehyde retrievals from S5P TROPOMI and from the QA4ECV project* by De Smedt et al., AMTD 2017**

Johannes Lampel

January 12, 2018

The manuscript *Algorithm Theoretical Baseline for formaldehyde retrievals from S5P TROPOMI and from the QA4ECV project* shows nicely the state of the TROPOMI HCHO product and relates it to other approaches.

I have a few short comments on technical aspects of the paper.

1. [Chance and Orphal, 2011] used the cross-section by [Meller and Moortgat, 2000] to rescale the higher resolved cross-section by [Cantrell et al., 1990], basically combining the advantages of both publications. Is there a specific reason why to use [Meller and Moortgat, 2000] alone?

2. At the lower end of the fitting interval $O_4$ absorption might play a role as reported in e.g. [Salow and Steiner, 1936] and shown for MAX-DOAS in [Lampel et al., 2017b]. How large is the impact on the retrieved HCHO SCDs? In this context: Maybe a typical (not smallest) residual could be included in figure 4 as an overlay in all subfigures to illustrate for a certain instrument the relation between noise and fitted structures, maybe using a comparable instrument or scaling the residual appropriately.

3. Figure 4 is missing a 'SCD' (if you want to call it like that) for the Ring.

4. At which temperature was the Ring spectrum calculated? For air-borne DOAS measurements the temperature dependence of the Ring effect can be significant (see [Volkamer et al., 2015]) and for ground-based measurements also two temperatures for the Ring spectrum were necessary (e.g. [Lampel et al., 2017a]) in order to avoid systematic structures of up to $5 \times 10^{-4}$. For ground-based measurements of HCHO, e.g. in the marine boundary layer, this is significant. The correction of this effect reduced also the HCHO/BrO cross-interferences for ground-based observations. I attached a plot for the absolute and relative difference of the Ring spectrum for a temperature difference of 30K (Ring(243K) - Ring(273K) ), see item 4. In Figure 4 you show that the OD attributed to the Ring is of the order of 4 %. A change of 2% of this signal is then of comparable magnitude as the OD attributed to HCHO.

[Figure]

5. Just curious: While vibrational Raman scattering (VRS) of liquid water plays up to 360nm no role (see [Peters et al., 2014] and references therein), does VRS of $N_2$ play a role in this spectral range (compare Figure 1)? For ground based measurements it is relatively hard to detect, but as residual spectra from satellite based retrievals keep looking better and better ... has this been tried? The largest part of it (the constant one in intensity space) will be compensated by the offset-polynomial, but then there is still the differential part, which should to some extend correlate with the overall signal of inelastic atmospheric scattering. Is this observed?

**References**

[Cantrell et al., 1990] Cantrell, C. A., Davidson, J. A., McDaniel, A. H., Shetter, R. E., and Calvert, J. G. (1990). Temperature-dependent formaldehyde cross sections in the near-ultraviolet spectral region. *The Journal of Physical Chemistry*, 94(10):3902–3908.

[Chance and Orphal, 2011] Chance, K. and Orphal, J. (2011). Revised ultraviolet absorption cross sections of $H_2CO$ for the HITRAN database. *Journal of Quantitative Spectroscopy and Radiative Transfer*, 112(9):1509 – 1510.

[Lampel et al., 2015] Lampel, J., Frieß, U., and Platt, U. (2015). The impact of vibrational raman scattering of air on doas measurements of atmospheric trace gases. *Atmospheric Measurement Techniques*, 8(9):3767–3787.

[Lampel et al., 2017a] Lampel, J., Pöhler, D., Polyansky, O. L., Kyuberis, A. A., Zobov, N. F., Tennyson, J., Lodi, L., Frieß, U., Wang, Y., Beirle, S., Platt, U., and Wagner, T. (2017a). Detection of water vapour absorption around 363 nm in measured atmospheric absorption spectra and its effect on doas evaluations. *Atmospheric Chemistry and Physics*, 17(2):1271–1295.

[Figure]

Figure 1: Figure 2 from [Lampel et al., 2015] showing the contribution to the overall intensity by rotational, vibrational and vibrational-rotational Raman scattering on $N_2$, $O_2$ and liquid water.

[Lampel et al., 2017b] Lampel, J., Zielcke, J., Schmitt, S., Pöhler, D., Frieß, U., Platt, U., and Wagner, T. (2017b). Detection of o$_4$ absorption around 328 nm and 419 nm in measured atmospheric absorption spectra. *Atmospheric Chemistry and Physics Discussions*, 2017:1–20.

[Meller and Moortgat, 2000] Meller, R. and Moortgat, G. K. (2000). Temperature dependence of the absorption cross sections of formaldehyde between 223K and 323K in the wavelength range 225-375 nm. *Journal of Geophysical Research: Atmospheres*, 105(D6):7089–7101.

[Peters et al., 2014] Peters, E., Wittrock, F., Richter, A., Alvarado, L. M. A., Rozanov, V. V., and Burrows, J. P. (2014). Liquid water absorption and scattering effects in doas retrievals over oceans. *Atmospheric Measurement Techniques*, 7(12):4203–4221.

[Salow and Steiner, 1936] Salow, H. and Steiner, W. (1936). Die durch Wechselwirkungskräfte bedingten Absorptionsspektra des Sauerstoffes 1. Die Absorptionsbanden des (O$_2$-O$_2$)-Moleküls. *Z. Physik*, 99:137–158.

[Volkamer et al., 2015] Volkamer, R., Baidar, S., Campos, T. L., Coburn, S., DiGangi, J. P., Dix, B., Eloranta, E. W., Koenig, T. K., Morley, B., Ortega, I., Pierce, B. R., Reeves, M., Sinreich, R., Wang, S., Zondlo, M. A., and Romashkin, P. A. (2015). Aircraft measurements of bro, io, glyoxal, no$_2$, h$_2$o, o$_2$-o$_2$ and aerosol extinction profiles in the tropics: comparison with aircraft-/ship-based in situ and lidar measurements. *Atmospheric Measurement Techniques*, 8(5):2121–2148.

---

## Referee Comment (RC2) · Anonymous Referee #2 · 25 Jan 2018

This work presents an extensive and detailed report of the retrieval algorithm of HCHO for TROPOMI on board of Sentinel-5. I believe that the manuscript is clearly written and well suits for the AMT so I recommend the publication with a few minor comments for clarification as follows:

L162: Please enlarge the size of tick and labels in Figures 3, 4. L208: "the assumption of a single effective light path" any supporting literature and previous studies would be highly appreciated for this sentence. L237-239: The BrO retrieved from the first fitting is used in the second fitting and it error may affect the retrieval of HCHO in the second fitting but the error analysis associated with this was not included in the manuscript.

[Figure]

L246: I guess that the retrieved HCHO from the second interval in Fig. 3 is adjusted considering the retrieved BrO from the first interval but is not clearly written in the manuscript. L280, L297-298: Is the sub-interval larger or shorter than a fitting window for each species, for example, HCHO? What if estimated shifts with the shorter sub-intervals differ within the fitting window, how would you apply this to the calibration? L299: It appears that the calibration of earth radiance follows the interpolation to the ir-radiance grid. But the sequence should be reversed I guess, otherwise, a possible shift can interfere interpolated radiance and needs to be corrected before the interpolation to the grids. L367: Can you take into account the variation of ozone columns with lati-tude for AMF using the US standard? Line 480-481: Using the measured radiance as reference spectrum instead of irradiance can reduce (or remove) row-dependent off-sets. Do we need to remove stripe patterns dependent on the row when the measured radiances are used as reference? Line 482-485, It would be appreciated if you can add some explanations about the cause for the latitudinal dependent offset. For example, Khokhar et al. (2005) suggested that the interference of O3 absorption may cause the latitudinal offset. Does this affect HCHO and BrO? Any quantitative information for the latitudinal offset? Does it also change in the two step fitting procedures with the wide and narrow fitting windows, respectively? In addition, if you used the irradiance as ref-erence it may need to account for the latitudinal variation of O3 and BrO absorption in the fitting.

Figure 8: It appears that not all M' is linear. So the question is how you include values for the nonlinear M' in the lookup table. Table 12: it is not clear what the use of x indicates.

---

## Author Comment (AC1) · 5 Mar 2018

We thank the reviewer for their positive and useful comments, and for their careful reading of the paper. We have addressed their questions as follows:

**Anonymous Referee #1**
De Smedt and co-authors describe in great detail the theoretical basis, methods and assumptions used in the operational processor of the Sentinel-5 Precursor TROPOspheric Monitoring Instrument (S5P TROPOMI). The algorithm described with great detail here benefits from strong heritage having the co-authors lead the development of formaldehyde retrievals using measurements from previous UV space sensors. Algorithm improvements developed during the Quality Assurance for Essential Climate Variables (QA4ECV) project, funded by the European Union, are also presented since they are the basis of future updates to the operational processor. The error budget for TROPOMI formaldehyde observations is derived and discussed in the context of the Copernicus requirements. Finally, validation methods and goals are discussed.

This paper should be the reference document for TROPOMI formaldehyde. Anyone using the operational product should read it to understand the meaning of the retrieved quantities, and their suitability to carry on scientific studies. The content is presented in a clear and sound way, it follows the logical steps of the algorithm and is well organized. The paper is ready to be published with minor changes that will only add to its great value. A set of recommendations to minimally expand the content of the paper is followed by some technical corrections.

**Section 1. Introduction.**
Some extra references should be added to support the statements describing formaldehyde chemistry in the atmosphere.

Ok. Done.

**Section 2.2 Algorithm Description**
Why are the HCHO Meller and Moorgat (2000) cross sections used instead of the more recent and intensity corrected Chance and Orphal (2011)?

The Chance and Orphal is based on the Cantrell et al., 1990 cross-sections, rescaled to match the Meller and Moortgat, 2000 cross-section.

Cantrell et al. offers a better spectral resolution (R = 0.011nm), but its absolute values are biased. With the 0.5 nm resolution of OMI and TROPOMI, we have chosen to use Meller and Moorgat (R = 0.025 nm) avoiding any handmade modification. The two datasets (Chance et Orphal and Meller and Moorgat) result in very consistent slant columns. As Chance and Orphal is the official HITRAN database, we will consider to switch the cross-sections, but this will not affect the results.

**Section 2.1.1 HCHO SCD retrieval**
Page 11, line 233. It will be valuable to show evidence of the reduction in the correlation between formaldehyde and bromine monoxide by using a two-step DOAS retrieval adding a new figure? Given the extension of the paper it is maybe not necessary, but it will be interesting to have it here.

For GOME2, we refer to De Smedt et al., 2012 (figure 7).

For OMI, I add a figure in this review (figure 1), based on the QA4ECV dataset for which the 2 fitting windows have been processed, the second without or with pre-fit of BrO (respectively WinB and WinC). The figure shows the standard deviation of the HCHO columns for 2005 in the remote Equatorial Pacific. The effect of pre fitting BrO columns in the small window (328.5-346 nm) can be seen by comparing WinB and WinC results (50% reduction of the noise level, from 1.5e16 to 1e16 molec.cm-2).

[Figure]

**Figure 1: Standard deviation of the OMI HCHO columns for 2005 in the remote Equatorial Pacific, as retrieved in WinA (328.5-359 nm), WinB (328.5-346 mn) or WinC (328.5-346 mn, with pre fit of BrO in WinA)**

Page 11, line 264. The text says "(3) possible row-dependent biases (stripes) are directly corrected owing to the use of one reference per detector row." Are irradiances not recorded for each detector row? If they are, as it is done with OMI, the reason for the removal of the stripes when using radiance reference should be other than just having an irradiance reference for each row.

We agree with this comment. The reason for the strong effect of using radiance as reference in reducing the stripes cannot be simply attributed to the use of each detector row. The reason is likely to be more complex and might rather be related to the observed improvement of the fits, involving a better cancelation of optical effects resulting in offsets (points (1) and (2)). We changed the text as:

"The main advantages of this approach are (1) an important reduction of the fit residuals (by up to 40%) mainly due to the cancellation of $O_3$ absorption and Ring effect present in both spectra; (2) the fitted slant columns are directly corrected for background offsets present in both spectra; (3) possible row-dependent biases (stripes) are greatly reduced by cancellation of small optical mismatches between radiance and irradiance optical channels; and (4) the sensitivity to instrument degradation affecting radiance measurements is reduced because these effects tend to cancel between the analyzed spectra and the references that are used"

Page 12, line 284. Equation 5 only shows the shift (Δi) but line 284 mentions that this approach allows compensating for stretch and shift errors. How correlated are those? Is it not possible to treat them separately?

The stretch can be seen as a wavelength-dependent shift, therefore shift and stretch are always related. In our approach, the overall stretch results from the use of several sub-windows in which shifts are fitted separately. Note that within each sub-window, a linear stretch around the central wavelength is also applied to avoid biases in the retrieved shift values.

**Section 2.2.2 Tropospheric air mass factor**

An evaluation of the effect of using only one atmospheric model (US Standard) for the calculation of altitude dependent air mass factors should be included. Ozone distribution can vary significantly between tropics, polar region and season.

This effect is considered in section 3.1.2, Table 8 (structural uncertainties). The two figures hereafter show the comparison of OMI HCHO AMF for one day (20050202), using different atmospheric profiles (US atmosphere, mid-latitude summer, mid-latitude winter). Our RT simulations show that this effect is relatively small for HCHO tropospheric AMFs. It can reach 10% at large SZA, but remains below 5% for classical observation conditions of HCHO. In the future, we will consider to add an additional dimension for O3 in the LUT of altitude resolved AMF, especially for sentinel-4 and its larger observation angles.

[Figure]

[Figure]

**Figure 2: Differences of OMI HCHO AMF for one day (20050202), using different atmospheric profiles (US atmosphere - mid-latitude summer (upper panel), US atmosphere - mid-latitude winter (lower panel)).**

Another subsection could be added to discuss the role of the surface reflectance properties to complete the description since all other AMF parameters have their own (LUT of altitude dependent AMFs, cloudy scenes, aerosols, and a priori vertical profile shapes).

The role of surface reflectance on HCHO AMFs is discussed in section 3.1.2. In section 2.2.2, our aim is to describe the algorithm, and for that, albedo is an auxiliary dataset to which we do not apply any modification.

**Section 2.2.3. Across-track and zonal reference sector correction**

Page 20, line 470. "The natural background level of HCHO is well estimated from chemistry model simulations of CH4". Actually there is evidence that models underpredict HCHO in the Pacific Ocean (http://onlinelibrary.wiley.com/doi/10.1002/2016JD026121/abstract). Add a sentence discussing this situation.

Thank you for this interesting paper. We added the reference in section 3.1.3 about the error on the reference sector correction. Our estimate of this error contribution is in the range of 1 to 2 x 10e15 molec./cm2. This is compatible with the differences between model and observations shown in figure 14 of Anderson et al., 2017. Please note that for this part, only the error on the HCHO total columns in the reference sector plays a role (not the profile). Note also that our reference sector is the Pacific Ocean, more remote from the continent than the sector presented in Anderson et al., that is closer to Indonesia, and therefore more influenced by continental sources.

As said above, despite the paper being fairly long, it could benefit of a plot showing the process of the background correction for one orbit illustrating the changes for each step.

We added this figure in section 2.2.3:

[Figure]

02 Feb. 2005   OMI HCHO Ns [x10$^{15}$ molec.cm$^{-2}$]

**Figure 3: Illustration of the across-track and zonal reference sector correction steps applied to one day of OMI HCHO slant columns (02/02/2005). The upper panel shows the uncorrected slant columns obtained using as DOAS reference spectrum the solar irradiance. The center panel shows the same slant columns after the first across-track correction step or when row averaged radiances selected in the Pacific Ocean are used as reference. The lower panel shows the final background corrected slant columns $\Delta N_s$ .**

**Section 3.1.1. Errors on the slant columns**
Please add O4 uncertainty to table 7.

We added 2% for an uncertainty of O4 cross-section. We tested the fit of an alternative cross-section (Herman et al.); and the fit of O4 in a dedicated window (339-364nm). The impact is pretty small on the absolute HCHO SCDs of one orbit.

**Section 3.1.2. Errors on air mass factors**

Page 26, Figure 9: Would it be possible to specify the geometry, surface pressure, and the rest of parameters kept fix in the calculation for each panel.

**Figure 9: AMF uncertainty related to profile shape, cloud pressure and surface albedo errors, as a function of different observation conditions. In all cases, we consider a nadir viewing and a solar zenith angle of 30°. By default, fixed values have been used. The surface pressure is 1063hPa, the albedo is 0.05, the effective cloud fraction is 0.5, and the profile height and cloud pressure are 795 hPa.**

**Page 26, Surface albedo:** Kleipool et al., surface climatology has a coarse resolution (0.5.x0.5.) compared with TROPOMI pixels. It would be interesting to incorporate in the error analysis the uncertainties associated with subpixel inhomogeneity in the Kleipool database or at least discuss them in the text.

It is true that the resolution of the Kleipool database is rather coarse in comparison to the size of Tropomi pixels. We hope to switch to a new database based on Tropomi measurements, as soon as possible (this has been added in Table 5). To our knowledge, there is no other LER database in the UV with a better spatial resolution.

In the error budget, it is shown that the impact of albedo uncertainty is significantly lower than profile shape uncertainty. This is because the albedo at UV wavelengths is generally small over regions where VOCs are emitted. Although significantly improved, the 1° resolution of the TM-5 can introduce uncertainty along coastlines for example. We have not done a detailed study using different model resolutions to quantify those errors. It is currently the subject of several studies. We added a note about the spatial resolution of auxiliary data in the paper.

**Page 27, Clouds and aerosols:** In section 2.2.2 "Tropospheric air mass factor" page 17, line 403 it is said that pixels with cloud fractions below 10% are considered clear-sky pixels "to avoid unnecessary error propagation through the retrievals" given the unstable cloud retrieval for such conditions. Under that assumption, how are the AMF errors due to cloud parameters calculated?

Our approach is to consider an error on the cloud fraction (0.05), even for very low cloud fraction up to 0. However, we neglect the error on the cloud pressure, since it is weighted by a low cloud fraction (equation 8).

**Page 27, Profile shape**: As for the surface climatology, would it be possible to estimate the uncertainty derived from subpixel model inhomogeneity given that model information is available in 1° grid and TROPOMI pixels can be as small as 7x7 km2.

See answer about albedo. It is currently difficult to provide a general and quantitative estimate of this uncertainty since we do not have access to global models running at a resolution finer than 1°x1°. This will be considered for future studies. However, we believe that our error estimate is conservative enough to include these finer effects.

**Section 4. Verification**

Building on the work by Lorente et al., 2017 was the AMF calculation tested using harmonized parameters.

The scattering weighting functions have been compared using harmonized parameters, showing excellent agreement. Each group used its own auxiliary data.

**Page 34, Table 12**: The number of xx in the Earth coverage column for MAX-DOAS and Direct Sun, should it not be similar given that most MAX-DOAS instruments can also carry on Direct Sun measurements?

This statement is incorrect. Currently only a minor fraction of the operated static MAX-DOAS instruments also feature direct-sun pointing capabilities (essentially Pandora systems).

Page 35, after paragraph devoted to MAX-DOAS could add a little paragraph describing Direct Sun capabilities.

Some MAX-DOAS systems (e.g. Pandora instruments) include a direct-sun pointing capability allowing for accurate total column measurements. It must be noted however that due to the faintness of the HCHO spectral signatures and the small geometrical enhancement in direct-sun geometry in comparison to MAX-DOAS, direct-sun measurements of HCHO are relatively difficult and not standard.

It will be nice to add a map of current ground-based measurements sites lined up for validation.

We rather refer to the ESA S5PVT project NIDFORVal, in which such a map is included (https://sentinel.esa.int/documents/247904/2474724/Sentinel-5P-Science-Validation-Implementation-Plan).

**Section 5.4 Satellite-satellite intercomparisons**
Page 36, line 861: For completeness about current and future instruments it will be good to add mentions to OMPS, GEMS, and TEMPO.

Ok, done.

**Technical Corrections:**
Page 1, line 33: "Its lifetime being of the order of a few hours, …" is grammatically incorrect. What about, "With its lifetime of the order of a few hours, HCHO concentrations in the boundary layer…"

Ok, Thanks.

Page 2, line 43: Would you consider to add Kaiser et al., 2017 to the list of inversion studies (https://doi.org/10.5194/acp-2017-1137)?

Done

Page 2, line 52: To complete the list of HCHO retrievals from LEO it should be added the ones using OMPS measurements (http://onlinelibrary.wiley.com/doi/10.1002/2015GL063204/abstract, https://www.atmos-meas-tech.net/9/2797/2016/).

Right, done.

Page 3, line 77: Is there any reference or link available to the S5P HCHO Level 2 Algorithm Theoretical Basis Document v1.0

I added the proper reference (De Smedt et al., 2016).

24x3h means a revisit time every 3 days, as it was the case for GOME-1 or SCIAMACHY. This is how it appears in the official requirements.

Ok. I have modified the figure.

Ok.

Done.

Correct.

Corrected.

Corrected.

Done

right

Corrected.

Corrected.

done

Corrected.

done

Page 50, line 1206: Remove sentence "A complete description of the level 2 data…" since it's a repetition of the sentence in line 1203.
Done. Thanks

---

## Author Comment (AC2) · 5 Mar 2018

The comment was uploaded in the form of a supplement:
https://www.atmos-meas-tech-discuss.net/amt-2017-393/amt-2017-393-AC2-supplement.pdf

---

## Author Comment (AC3) · 5 Mar 2018

We would like to thank J. Lampel for his careful reading of the paper. We will certainly consider to apply the proposed technical tests.

*Short comment: Algorithm Theoretical Baseline for formaldehyde retrievals from S5P TROPOMI and from the QA4ECV project by De Smedt et al., AMTD 2017, Johannes Lampel*

*The manuscript Algorithm Theoretical Baseline for formaldehyde retrievals from S5P TROPOMI and from the QA4ECV project shows nicely the state of the TROPOMI HCHO product and relates it to other approaches. I have a few short comments on technical aspects of the paper.*

1. *[Chance and Orphal, 2011] used the cross-section by [Meller and Moortgat, 2000] to rescale the higher resolved cross-section by [Cantrell et al., 1990], basically combining the advantages of both publications. Is there a specific reason why to use [Meller and Moortgat, 2000] alone?*

The Chance and Orphal is based on the Cantrell et al., 1990 cross-sections, rescaled to match the Meller and Moortgat, 2000 cross-section. Cantrell et al. offers a better spectral resolution (R = 0.011nm), but its absolute values are biased. With the 0.5 nm resolution of OMI and TROPOMI, we have chosen to use Meller and Moorgat (R = 0.025 nm) avoiding any handmade modification. The two datasets (Chance et Orphal and Meller and Moorgat) result in very consistent slant columns. As Chance and Orphal is the official HITRAN database, we will consider to switch the cross-sections, but this will not affect the results.

2. *At the lower end of the fitting interval O4 absorption might play a role as reported in e.g. [Salow and Steiner, 1936] and shown for MAX-DOAS in [Lampel et al., 2017b]. How large is the impact on the retrieved HCHO SCDs? In this context: Maybe a typical (not smallest) residual could be included in figure 4 as an overlay in all subfigures to illustrate for a certain instrument the relation between noise and fitted structures, maybe using a comparable instrument or scaling the residual appropriately.*

We have not tested O4 absorption cross-section dataset including a band around 328 nm. However, according to Lampel et al 2017b, the maximum of this band is about 5 times smaller than the band at 342 nm, which itself is significantly weaker than the absorption at 360nm. We do not expect a strong impact on satellite observations.

3. *Figure 4 is missing a 'SCD' (if you want to call it like that) for the Ring.*

Thanks. The legend has been modified;  8% of inelastic scattering has been considered.

4. *At which temperature was the Ring spectrum calculated? For air-borne DOAS measurements the temperature dependence of the Ring effect can be significant (see [Volkamer et al., 2015]) and for ground-based measurements also two temperatures for the Ring spectrum were necessary (e.g. [Lampel et al., 2017a]) in order to avoid systematic structures of up to $5 \times 10^{-4}$. For ground-based measurements of HCHO, e.g. in the marine boundary layer, this is significant. The correction of this effect reduced also the HCHO/BrO cross-interferences for ground-based observations. I attached a plot for the absolute and relative difference of the Ring spectrum for a temperature difference of 30K (Ring(243K) - Ring(273K) ), see item 4. In Figure 4 you show that the OD attributed to the Ring is of the order of 4 %. A change of 2% of this signal is then of comparable magnitude as the OD attributed to HCHO.*

We have taken 350K. Thank you for the suggestion. We will surely test the impact for satellite retrievals of HCHO.

5. *Just curious: While vibrational Raman scattering (VRS) of liquid water plays up to 360nm no role (see [Peters et al., 2014] and references therein), does VRS of N2 play a role in this spectral range*

(compare Figure 1)? For ground based measurements it is relatively hard to detect, but as residual spectra from satellite based retrievals keep looking better and better ... has this been tried? The largest part of it (the constant one in intensity space) will be compensated by the offset-polynomial, but then there is still the differential part, which should to some extend correlate with the overall signal of inelastic atmospheric scattering. Is this observed?

This has not been tested.